# Background optic flow modulates responses of multiple descending interneurons to object motion in locusts

**Sinan Zhang**[ID]¤*, **John R. Gray**

Department of Biology, College of Arts and Science, University of Saskatchewan, Saskatoon, Saskatchewan, Canada

¤ Current address: Department of Medical Genetics, Cumming School of Medicine, University of Calgary, Calgary, Alberta, Canada
* sinan.zhang@ucalgary.ca

## Abstract

Animals flying within natural environments are constantly challenged with complex visual information. Therefore, it is necessary to understand the impact of the visual background on the motion detection system. Locusts possess a well-identified looming detection pathway, comprising the lobula giant movement detector (LGMD) and the descending contralateral movement detector (DCMD). The LGMD/DCMD pathway responds preferably to objects on a collision course, and the response of this pathway is affected by the background complexity. However, multiple other neurons are also responsive to looming stimuli. In this study, we presented looming stimuli against different visual backgrounds to a rigidly-tethered locust, and simultaneously recorded the neural activity with a multichannel electrode. We found that the number of spike-sorted units that responded to looms was not affected by the visual background. However, the peak times of these units were delayed, and the rise phase was shortened in the presence of a flow field background. Dynamic factor analysis (DFA) revealed that fewer types of common trends were present among the units responding to looming stimuli against the flow field background, and the response begin time was delayed among the common trends as well. These results suggest that background complexity affects the response of multiple motion-sensitive neurons, yet the animal is still capable of responding to potentially hazardous visual stimuli.

## Introduction

Successful navigation in the natural environment requires animals to rapidly perceive and extract important sensory cues, usually from a complex, "noisy" background. For example, the self-motion of animals generates optic flow, which is defined as the pattern of image shifts caused by the motion of the observer [1]. Although animals

**Data availability statement:** All relevant raw data are available from the Figshare database (https://doi.org/10.6084/m9.figshare.29437058.v2).

**Funding:** This study was funded by the Natural Sciences and Engineering Research Council of Canada (Award RGPIN-2019-03983). The funder had no role in study design, data collection and analysis, decision to publish, or preparation of the manuscript.

**Competing interests:** The authors have declared that no competing interests exist.

can use optic flow to control locomotion speed and correct for deviations in flight orientation [2–6], the presence of optic flow also makes it more difficult to observe other moving objects, such as an approaching predator. Successful detection and discrimination between different types of visual motion are critical to an animal's survival.

In the mouse retina, four subtypes of direction-sensitive ganglion cells (DSGC) are involved in the encoding of optic flow [7]. Each subtype of DSGC is selectively sensitive to optic flow along a specific translatory orientation, including forward, backward, up, and down. Combinations of these subtypes can also encode rotatory degrees of freedom. Flying animals, however, manoeuvre in additional degrees of freedom, including but not limited to vertical translation and roll, neither of which is commonly executed by land animals. Therefore, flying animals need to process more complicated optic flow parameters. With a relatively tractable nervous system and the capability to perform complex behaviours, insects are ideal subjects to study the neural coding of complex visual backgrounds. In flies, six types of direction-sensitive neurons are involved in the computation of the optic flow direction [8].

The migratory locust, *Locusta migratoria*, is another classic subject for studying visual processing. Migratory locusts are polyphenic, transitioning between solitary and gregarious states. In the gregarious state, locusts can form large swarms, comprised of thousands of individuals, and fly $\sim$3 meters·s$^{-1}$ [9]. With conspecifics moving around at various velocities from different angles, along with the flow field generated from self-motion, locusts flying in a swarm can still avoid collision with conspecifics [10] and predatory birds [11]. In locusts, an established neural pathway, including the lobula giant movement detector (LGMD) and its postsynaptic partner, the descending contralateral movement detector (DCMD), has been identified to respond preferably to objects on a collision course [12–15].

The LGMD/DCMD pathway is considered to play an important role in generating collision avoidance behaviours. Each LGMD receives input from an entire compound eye [16]. When presented with an approaching object, the LGMD responds with increased firing rate, which peaks near the projected time of collision (TOC) and then decays [17–22]. Each LGMD spike generates a spike in the DCMD [23], which synapses with the motor center related to avoidance behaviours [24]. Different phases of responses recorded in the LGMD/DCMD pathway have been associated with critical timing of avoidance behaviours, such as jumping [25] and flight steering [26].

The addition of a visual background affects the response of the LGMD/DCMD pathway to looming objects. In the LGMD, the number of spikes evoked by small looming objects is inhibited by the presence of large-field optic flow [14,27,28]. In the DCMD, although the general peak response is largely invariant between background types, the flow field still caused a lower peak firing rate, delayed peak time, shorter rise phase, and longer fall phase [29].

Previously, we investigated putative population coding of motion-sensitive neurons in response to objects moving against a simple white background [30]. To further understand how motion-sensitive neural ensembles behave in an environment that emulates elements of a more natural environment, we presented a looming stimulus approaching against one of three different backgrounds: solid white, solid grey in

the lower visual field, and a grey and white flow field that represented forward motion through space at an average locust flight speed of 3 m·s⁻¹. These stimuli were presented to rigidly-tethered locusts to allow recording the simultaneous activity of multiple descending interneurons. Based on previous reports, we tested the following hypotheses: 1) Responses of sorted units can be categorized based on their firing rate properties, 2) The distribution of response categories differs in the presence of different backgrounds, and 3) The firing rate properties of units within response categories are affected by the background. Overall, most spike-sorted units responded to looming stimuli regardless of background types, but the distribution of different types of responses was affected. For individual units that peaked near TOC, as well as common trends that peaked near TOC, the peak time and/or the response begin time was delayed by the flow field background.

## Materials and methods

### Animals

Twenty-one adult male locusts (*Locusta migratoria*) were chosen from a colony maintained at the University of Saskatchewan, Saskatoon, Canada, ensuring that they were at least 3 weeks past the imaginal moult. These locusts were fed with a diet consisting of wheat grass and bran and were subjected to a regular light-dark cycle of 12 hours of light and 12 hours of darkness at an approximate temperature of 30 °C. To maintain the gregarious state, the locust colony was kept in a crowded condition [31].

### Preparation

During the light cycle, the locusts were carefully removed from the colony and transferred to a wire mesh container. To provide both light and warmth, an incandescent lamp was placed on top of the container. The experiments were conducted at a controlled room temperature of approximately 25 °C.

To prepare the locust for the experiments, the legs were first removed, and a rigid tether was then securely attached to the ventral surface of the thorax using melted beeswax. A small portion of the ventral cervical cuticle was removed to expose the paired connectives of the ventral nerve anterior to the prothoracic ganglion. The prepared locust was then transferred to the stimulus arena [32] (Fig 1A). A silver wire hook electrode was used to lift and stabilize the left ventral nerve cord. Subsequently, a sharp glass electrode was carefully inserted into the protective sheath surrounding the left nerve cord, creating a small opening. A twisted wire tetrode, fabricated following an established protocol [33], was then inserted into the left nerve cord anterior to the hook electrode (Fig 1B). The tetrode consisted of four wires (diameter = 12.7 µm) (Model R0800, Sandvik-Kanthal Precision Technology, Hallstahammar, Sweden) that were twisted together and fused at one end, while the other end was separated and soldered to four channels of an adapter. The fused end of the tetrode was secured within a capillary tube for stability. A micromanipulator was used to maneuver the capillary tube, lowering the fused end of the tetrode into the nerve cord, while the other end (adapter) was connected to a differential amplifier (Model 1700, A-M Systems, United States). Finally, a 0.5-mm silver ground wire (MilliporeSigma Canada, Oakville, ON, Canada) was inserted into the abdomen. We used two tetrodes for the entire dataset. Therefore, tetrodes were reused between preparations. At the time of fabrication, the impedance was measured for both tetrodes. The impedance within each channel was < 3 megaohm, and the impedance between channels was > 5 megaohms. The tip that was inserted into the nerve cord was trimmed before each prep. Since the tip was not coated, the trimming would not have affected the impedance.

Neural responses were observed while waving a hand to verify a high signal-to-noise ratio. A mixture of mineral oil and Vaseline was then applied around the electrodes and nerve cords to insulate the recordings from the hemolymph and prevent desiccation. The entire setup was rotated 180°, placing the locust in a dorsal-side-up position at the center of the rear projection dome screen (Fig 1C). The longitudinal axis of the locust was aligned parallel to the apex of the dome, with a distance of 12 cm between the locust's eye and the screen. In this orientation, 0° represented the front of the locust, 180° represented the direction directly behind, and +90° represented a perpendicular angle to the right longitudinal axis of the locust's body. Negative perpendicular angles were to the left of the locust.

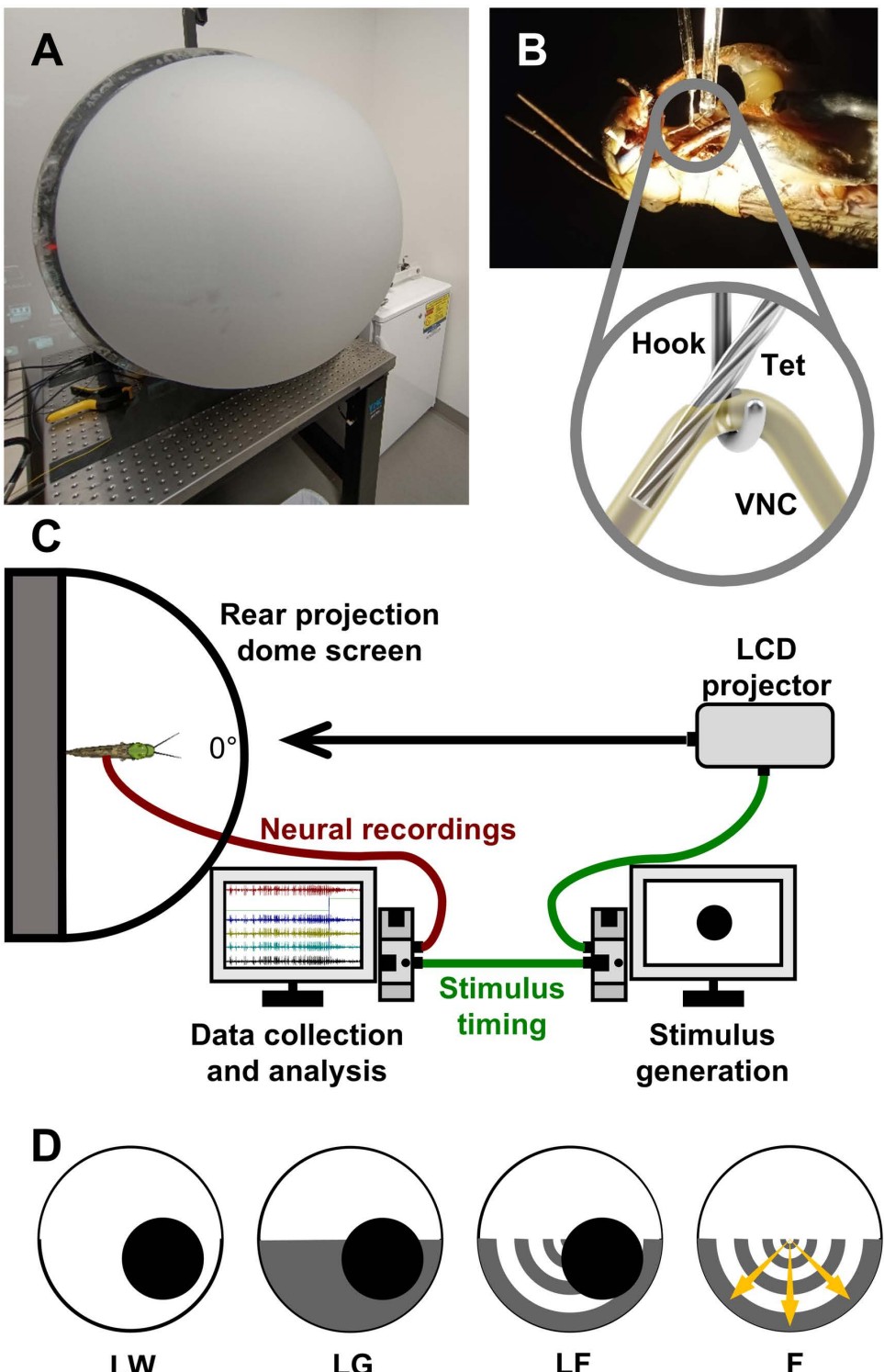

**Fig 1. Experimental setup and visual stimuli. (A)** The stimulus arena with the dome screen closed. **(B)** The locust preparation with tetrode inserted (inset). **(C)** Diagram of the experimental setup. Visual stimuli were rendered and projected onto the dome screen while neural activity was recorded simultaneously. A 5-V pulse was used to align the neural recordings with stimulus timing. 0° represents the orientation relative to the head of the locust (locust not to scale). **(D)** Visual stimuli used in the experiment. The first three visual stimuli included a 7-cm-diameter black disc approaching at 300

cm·s⁻¹ from the bottom right at a 45° angle, against three visual backgrounds: white (LW), half-white and half-grey (LG), and flow field (LF). The last visual stimulus was composed of a flow field only **(F)**. Yellow arrows within F indicate the direction of flow field motion, which was the same for LF. The thickness of the arrows and grey arches increased as the flow field expanded. Each visual stimulus was presented 5 times, and the order of all 20 visual stimuli was randomized for each locust.

### Visual stimulus

Visual stimuli were generated using a Python-based program EVSG [34] and incorporated correctional factors to account for the curvature of the dome screen. The resolution of the stimuli was set at 1024 × 768 pixels. Each pixel on the screen corresponded to approximately 0.7 mm, resulting in a subtense angle of approximately 0.4°, which is well below the angular resolution of locusts' eyes (1°) [35].

Real-time rendering of the images was performed using a GeForce RTX 3080 video card (NVIDIA Corporation, Santa Clara, United States), achieving a frame rate of over 150 frames per second. The images were then projected onto the dome screen using an InFocus DepthQ projector (InFocus, Portland, United States) with the colour wheel removed. The frame rate of the projections exceeded the flicker fusion frequency of locusts' eyes, which is around 66 Hz [36].

A 5V pulse was generated at the projected time of collision (TOC), which was automatically calculated based on the motion trajectory of the visual stimuli. This pulse was used to align different trials. Given the curvature of the dome screen and the position of the locust relative to the apex, we were able to render the final frame such that the visual subtense angle of the object was 180°, which is the point of collision.

In all visual stimuli, the backgrounds were divided vertically into two equal halves, simulating a horizon line positioned at 0° elevation. The top half was rendered in solid white (R = 0, G = 0, B = 0), while the bottom half was either solid white (LW), solid grey (LG, R = 100, G = 100, B = 100), or a flow field (LF), composed of grey concentric arches (R = 100, G = 100, B = 100) expanding outward from the center of the dome below the horizon line. The solid grey background was designed to test the putative effects of contrast between the lower and upper visual fields. The arches expanded at ~40.5°·s⁻¹ across the surface of the dome, which represented forward motion of 300 cm·s⁻¹, which is the average flight speed of a locust. At the center of the dome, the arch subtended 6.8° of the visual field, and at the periphery it subtended ~13.5°. The Michelson contrast ratio between the grey and the white backgrounds was 0.87 (Fig 1D). Against each background, a 7-cm diameter black disc (R = 255, G = 255, B = 255) approached the locust at a velocity of 300 cm·s⁻¹ (looming) from the bottom right at 45° azimuth and −45° elevation. The trajectory of the looming disc was designed so that it remained within the lower half of the dome screen for the majority of its motion. Additionally, a fourth visual stimulus (F) consisted of only the flow field below the horizon line.

Previous studies used flow field backgrounds that consisted of vertical bars that expanded across the entire visual field [29,37], or squares expanding from the center of the screen [20]. The type of flow field used here was designed to emulate the natural visual environment when the locust is flying. Since flying animals are closer to the ground than the sky, the objects on the ground would appear faster and more conspicuous. Therefore, the optic flow in the lower half of the visual field is more prominent than that in the top half. By dividing the visual background vertically into two halves and showing the flow field in the bottom half only, we presented a simplified background that simulates a horizon line. Preliminary testing indicated that there was no effect of a flow field on looming responses when the object approached from 45° azimuth at 0° elevation. For this trajectory, the top half of the object approached within the white background, which likely attenuated any putative effects of a flow field [29], regardless of flow field contrast or velocity. Therefore, we chose a looming stimulus that moved upwards from −45° elevation to ensure the disc was within the flow field except for the final 20.2 ms of each approach. When presented with looming stimuli approaching within different areas across the receptive field, some parameters of the DCMD response are different, yet the shapes of the PSTH remain largely unaffected [38,39]. We found a similar effect among the other motion-sensitive units and common trends extracted from these units. We collected similar multichannel data in an independent dataset that tested different hypotheses with respect to object motion at different

elevation angles (manuscript in review). From this data, we observed that the number of responsive units and the number of common trends, representing the types of responses among all responsive units, were consistent between looming at 0° elevation and −45° elevation.

Locusts display motion dazzle. When presented with a two-tone looming object, in which the top half is darker than the background while the bottom half is lighter, the DCMD responds less than the top (dark) half alone [40]. However, the flow field we used here was composed of white and grey circles, both lighter than the black object (Michelson contrast ratio between black and white = 0.98, Michelson contrast ratio between black and grey = 0.80). Therefore, the motion dazzle was likely not present in our stimuli.

Each visual stimulus was presented 5 times, resulting in a total of 20 presentations, the order of which was randomized for each locust. To prevent neural habituation, the visual stimuli were presented with a 3-minute interval between each presentation. Additionally, a direct loom was presented before and after the sequence of the 20 presentations to ensure that no attenuation of the neural responses occurred during the length of the experiment.

## Data acquisition

The neural signals from all 5 channels, including 1 channel from the silver hook electrode and 4 channels from the twisted wire tetrode, were amplified using a differential amplifier (Model 1700, A-M Systems, Sequim, United States) with a high pass filter at 300 Hz, a low pass filter at 5 kHz, and a gain of 100×. These amplified neural signals, along with the stimulus pulse, were then digitized using a USB data acquisition board (DT9818-OEM, TechmaTron Instrument Inc., Laval, Canada) and recorded at a sampling rate of 25 kHz per channel using DataView version 11 (W.J. Heitler, University of St Andrews, St Andrews, Scotland).

## Spike sorting

For each locust, all 20 raw recordings were merged chronologically into a single file and then filtered using a Finite Impulse Response (FIR) filter with a bandpass range of 500–2000 Hz and a Blackman window type using DataView. The filtered data from the four channels of the tetrode were exported to Offline Sorter version 4.6 (Plexon Inc., Dallas, United States) for waveform detection and spike sorting. The waveform detection threshold was set to three times the standard deviation over the mean of each channel. The detection time was set such that each waveform was contained within a 2-ms window and then aligned to the largest peak on either side. Spike sorting was performed using a semi-automatic method based on the T-Dist E-M algorithm [41]. The degree-of-freedom (DOF) multiplier was set to 3, and the initial number of units was set to 26. Subsequently, manual discrimination was based on the shape and amplitude of the waveforms for final unit sorting. A Multivariate Analysis of Variance (MANOVA) was used to compare the distribution of the sorted units in the 3D principal component (PC) space. We used the built-in MANOVA test provided by the Offline Sorter software to statistically validate the separation of our sorted units. This test compares the distributions of spike data points in the 3D feature space, defined by the first three principal components. The independent variable for the test is the assigned unit (cluster), and the dependent variables are the feature vectors (i.e., the PC scores) for each individual waveform within those units. Fig 2A shows raw recordings and sorted units from one trial of an object approaching against a white background, whereas Fig 2B and 2C show PC clusters and overlapping waveforms from units sorted from the merged file that contains all 20 stimulus trials for the same locust.

## Spike train analysis

The spike times of spike-sorted units were exported to NeuroExplorer Version 5 (Plexon, Dallas, United States) for further analysis. Peristimulus time histograms (PSTHs) were generated using a 1-ms bin width and smoothed with a 50-ms Gaussian filter. All trials were aligned to the projected time of collision (TOC). For trials involving a looming disc, a 2.2-second window was used, starting 2 seconds before TOC, and ending 0.2 seconds after TOC. For trials with the flow field only, a 10-second window was employed, spanning 5 seconds before the flow field onset and ending 5 seconds after.

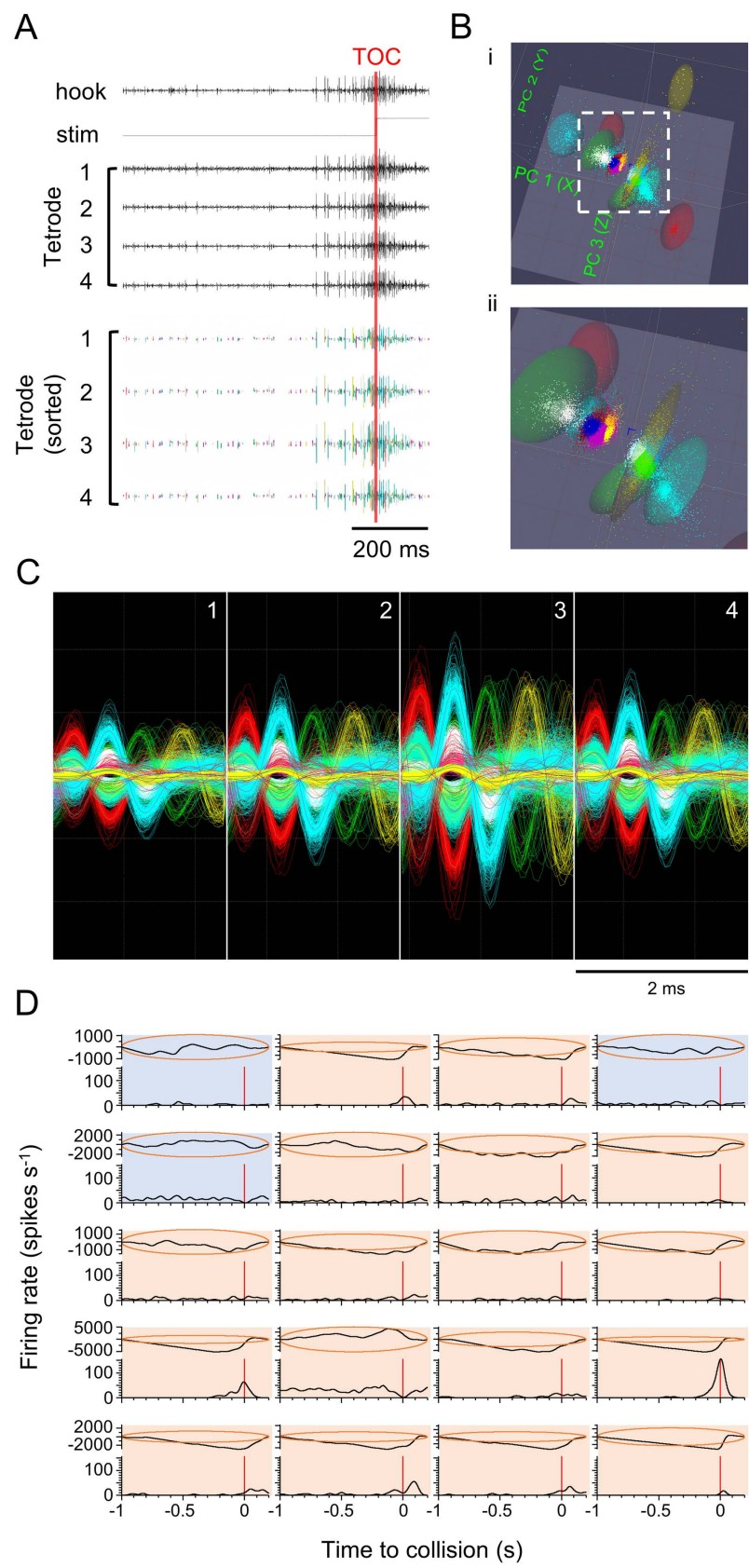

**Fig 2. Spike sorting and identification of responsive units. (A)** Raw recording and sorted units from one trial of an object approaching against a white background. The top 6 channels show raw recordings from the hook electrode, the stimulus event pulse (stim), and each channel of the tetrode. Waveforms (bottom) were detected using average plus three times the standard deviation as the threshold, and sorted based on the cluster variance in the 3D principal component (PC) space. Spike sorting results from the same recording time window (tetrode sorted) are shown underneath the raw recordings. Each colour represents a different unit. **(B)** The distribution of all 20 spike-sorted units in the 3D PC space, with (B.i) showing all units, and (B.ii) showing a zoomed-in view of the unit clusters close to the center, as labeled by the dashed white box in (i). The colour coding of each sorted unit is the same as in A. **(C)** Overlapped waveforms of all 20 units from the same locust, recorded from all four channels of the tetrode, as numbered on the top right corner of each panel. The colour coding of each sorted unit is the same as in A and B. **(D)** Identification of responsive units. For each spike-sorted unit, the peristimulus time histogram (PSTH), representing the change of firing rate during the stimulus presentation, is shown at the bottom of each block. The cumulative sum and 99% confidence level (represented by the dark orange ellipses) are plotted on the top half of each block. If the cumulative sum (represented by the black line inside the ellipse) touched or extended outside the ellipse (plots with a light orange background), the unit was determined to have responded to the stimulus. If the cumulative sum remained within the ellipse for the entire duration (plots with a light blue background), the unit was determined to have not responded. Note that not all units generated spikes during this stimulus presentation. Inactive and non-responsive units were removed from subsequent analyses.

To evaluate the responsiveness of individual units, cumulative sum plots were created along with an ellipse representing the 99% confidence level (Fig 2D) [42,43]. The cumulative sum was calculated in NeuroExplorer using the following algorithm:

$$\text{For bin 1}: \ cs(1) \ = \ bc(1) \ - \ A;$$

$$\text{For bin 2}: \ cs(2) \ = \ bc(1) \ + \ bc(2) \ - \ A \times 2;$$

$$\text{For bin 3}: \ cs(3) \ = \ bc(1) \ + \ bc(2) \ + \ bc(3) \ - \ A \times 3;$$

$$\dots$$

In this algorithm, $cs(n)$ represents the cumulative sum of the $n$-th bin, $bc(n)$ represents the spike bin count of the $n$-th bin, and A represents the average of all bin counts. If the cumulative sum remained within the 99% confidence level ellipse, it indicated that the firing rate of the corresponding unit did not exhibit a significant change during the stimulus presentation, implying that the unit did not respond to the visual stimulus. Conversely, if the cumulative sum touched or extended beyond the ellipse, it was considered responsive to the visual stimuli [42–44]. Units that did not exhibit a response were excluded from subsequent analyses.

## Dimensionality reduction

The responses of individual units were first examined to determine how many types of visual stimuli each unit responded to. Between different animals, the same neuron could have been recorded multiple times. Therefore, we pooled the responsive units across all animals based on the visual stimulus. For each stimulus, the response types were initially categorized by observing the peristimulus time histogram (PSTHs) of individual units. The categorization was adopted from previous studies [42,44]. Firing parameters, including the peak time relative to TOC ($P_t$), peak firing rate amplitude ($P_a$), rise phase (from the time when the unit firing rate exceeded the upper 95% confidence interval to the peak time), and decay phase (from the peak time to when the firing rate decayed below 15% of the peak firing rate) [21,45], were measured from the PSTHs of those units that displayed a clear peak, and compared between different backgrounds. Due to the nature of multichannel recording, it was not possible to record unambiguously from the same neurons in each locust and, therefore, it was necessary to reduce the dimensionality of the data to identify putative trends across the entire dataset [42].

Dynamic Factor Analysis (DFA) was then used to extract common trends from the responsive units for each visual stimulus [42,46]. In this model, a set of $n$ observed time series (**y**) (units) is explained by a set of $m$ hidden random walks (**x**) (common trends) through linear combination, represented by factor loadings (**Z**) and offsets (**a**). The model equations can be written as follows:

$$\mathbf{y}_t = \mathbf{Z} \times \mathbf{x}_t + \mathbf{a} + \mathbf{v}_t, \text{ where } \mathbf{v}_t \sim \text{MVN}(0, \mathbf{R})$$

$$\mathbf{x}_t = \mathbf{x}_{(t-1)} + \mathbf{w}_t, \text{ where } \mathbf{w}_t \sim \text{MVN}(0, \mathbf{Q})$$

In the model, the error terms of hidden processes (**x**) (common trends) and observations (**y**) (units) were represented by $\mathbf{w}_t$ and $\mathbf{v}_t$, respectively. Both $\mathbf{w}_t$ and $\mathbf{v}_t$ follow a multivariate normal distribution (MVN), with a mean of 0 and covariate matrices of **R** and **Q**, respectively. To identify the model, certain constraints were applied: the first $m$ elements of **a** were set to 0, $z_{i,j}$ was set to 0 if $j>i$ for the first $m$-1 rows of **Z**, and **Q** was set to the identity matrix $\mathbf{I}_m$.

DFA was performed using the MARSS package v3.11.4 [47] in R 4.1.3 (R Core Team, 2022), using the Broyden–Fletcher–Goldfarb–Shanno (BFGS) method. The model quality was evaluated using the Akaike Information Criterion with a correction for small sample size (AICc) [48]. For each visual stimulus, DFA was iteratively performed, starting with one common trend. The number of common trends gradually increased until AICc began to increase, and the model with the lowest AICc was selected as the optimal model.

The extracted common trends were further analyzed. The factor loadings (Z) of each common trend were used to rectify any potentially reversed trends and scale the common trends. According to the DFA model, $\mathbf{y}_t = \mathbf{Z} \times \mathbf{x}_t + \mathbf{a} + \mathbf{v}_t$, for the $i$-th common trend $((x^T)_i)$ and the corresponding factor loadings $(Z_i)$, if $(x^T)_i$ was multiplied by a scale factor, while $Z_i$ was divided by the same scale factor, the overall model is not affected. The sign of the scale factor was determined by the average of $Z_i$, while the value was determined by the largest absolute value among all elements within $Z_i$. If the average of $Z_i$ was negative, it indicates that the common trend contributes negatively to most unit responses (observations), and therefore, reversing the sign would make the common trend consistent with the majority of units. The value was chosen to scale all factor loadings to $(-1, 1)$, and therefore, to make the scale of all common trends comparable to each other. For common trends showing clear peaks, various parameters, such as peak time, rise phase, and decay phase, were measured from the PSTHs. These parameters were then compared between different backgrounds.

## Statistical analysis

Statistical analyses were executed using SigmaPlot version 12.5 (Systat Software, San Jose, USA) or R 4.1.3 (R Core Team, 2022), and plotted using SigmaPlot. Prior to applying specific tests, datasets were preliminarily assessed for normal distribution (Shapiro-Wilk test) and homoscedasticity. Parametric data was articulated through arithmetic mean and standard deviation, and plotted with a column graph. Nonparametric data was articulated through median and quartiles, and plotted with a box plot. Response parameters were compared through either one-way Analysis of Variance (ANOVA) (for parametric data) or Kruskal-Wallis One Way Analysis of Variance on Ranks (for nonparametric data). Variables with an inherent relationship were compared using either one-way repeated measure (RM) ANOVA (for parametric data) or Friedman Repeated Measures Analysis of Variance on Ranks (for nonparametric data). Two-way comparisons were executed utilizing two-way RM ANOVA and Holm-Sidak post-hoc test. All statistical examinations were two-tailed, and the significance level ($\alpha$) was determined to be 0.05. Power analysis for ANOVA used data from previous studies to estimate the minimal detectable difference in means, the expected standard deviation of residuals, the number of groups, and group size. The power was $>0.8$ for ANOVA and Chi-square tests.

## Results

### Unit discrimination

All twenty trials from each animal were chronologically merged for spike discrimination and unit sorting. Utilizing a positive threshold of three times the standard deviation above the mean, we detected 571,515 spikes across all twenty-one animals (median = 25,637 spikes per locust, 1st quartile = 23,692 spikes per locust, 3rd quartile = 29,090.5 spikes per locust). From these spikes, we spike-sorted 352 distinct units (mean = 16.8 units per locust, standard deviation = 3.4 units per locust). Multivariate Analysis of Variance (MANOVA) verified the statistical distinctiveness of these units in three-dimensional PC space ($p < 0.001$). A comprehensive summary of the spike-sorted spikes and units for all animals is provided in S1 Table.

### Unit responses

S2 Table summarizes the number of units that responded to each stimulus type. Out of the total 352 spike-sorted units, 248 units (70%) exhibited responses to looming stimuli against a white background (LW), 238 units (68%) responded to looming stimuli against a half-white-half-grey background (LG), 242 units (69%) responded to looming stimuli against a flow field background (LF), 222 units (63%) responded to the flow field only (F), and 28 units (8%) did not respond to any of the visual stimuli. One-way ANOVA revealed that there was no significant difference in the number of units responding to looming stimuli against different backgrounds ($F_2 = 0.14$, $p = 0.87$), nor was there a significant difference in the percentage of responsive units from each animal (One-way ANOVA, $F_2 = 0.49$, $p = 0.61$) (Fig 3).

However, it is worth noting that the responses of these units were not exclusive to a single background. A unit could respond to looming stimuli against one or more visual backgrounds. Fig 4 illustrates the distribution of the percentage of different types of looming stimuli that the units responded to. Among the total 352 units, 63 units (15%) responded to only one type of looming stimulus (1 Loom), among which 20 units responded to LW, 18 units responded to LG, and 25 units responded to LF. 42 units (12%) responded to two out of the three looming stimuli (2 Looms), among which 19 units responded to LW & LG, 15 units responded to LW & LF, and 8 units responded to LG & LF. 194 units (55%) responded to

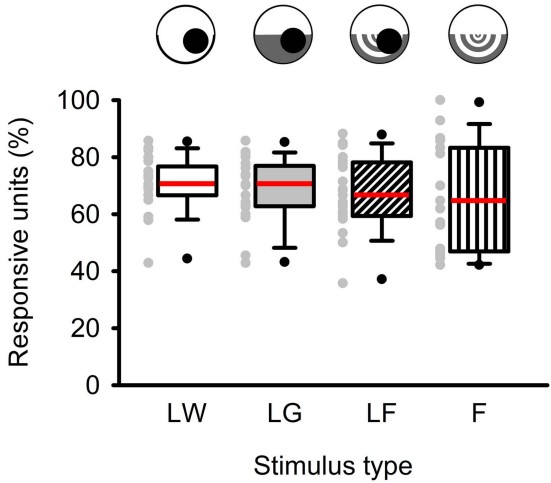

**Fig 3. Percent of units that responded to each type of visual stimulus.** Within each box, the red horizontal line represents the median, while the lower and upper boundaries, the whiskers, and the dots represent the 25/75%, 10/90%, and 5/95% percentiles, respectively. Grey symbols to the left of each box are the individual data points, each from one locust (N = 21). One-way repeated measure (RM) ANOVA revealed that there was no statistical difference between the number of units that responded to different types of visual stimuli. LW = looming against a white background, LG = looming against a half-white and half-grey background, LF = looming against a flow field, F = flow field only.

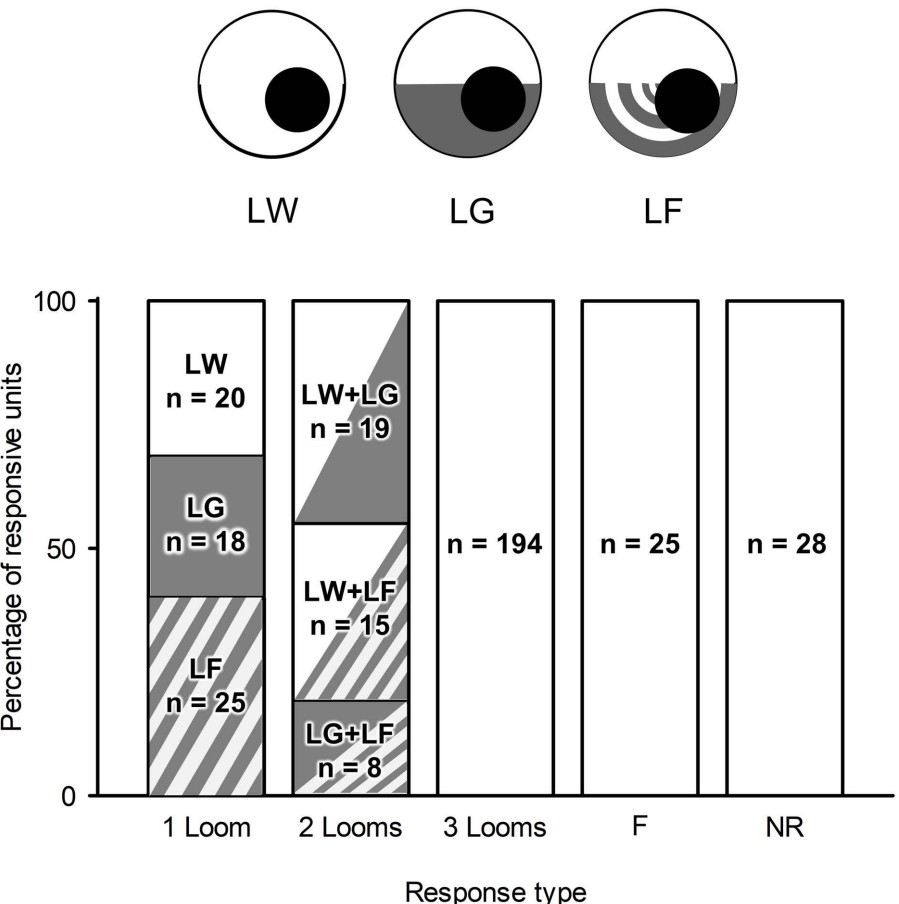

**Fig 4. Percentage of responsive units based on stimulus properties.** Each column represents the number of units (n) that responded to looming stimuli flow field only, or none. Out of all 352 spike-sorted units, 63 units (18%) responded to looming stimuli against only one background (1 Loom), 42 units (12%) responded to looming stimuli against two backgrounds (2 Looms), and 194 units (55%) responded to looming stimuli against all three backgrounds (3 Looms). See text for details. Overall, the background type did not affect how many units responded to the looming stimulus. NR = no response.

looming stimuli against all three backgrounds (3 Looms). Additionally, 25 units responded exclusively to the flow field (F) and none of the looming stimuli, and 28 units did not respond to any stimuli (NR).

Overall, the number of units responding to looming stimuli was relatively similar across all three backgrounds, suggesting that the visual background did not affect the number of units responding within a putative population.

## Categorization of unit response

Previous studies showed the classification of units based on firing rate properties [42,44]. We identified similar categories based on the trends of firing rate changes over time in response to a similar stimulus (looming object approaching against a white background). Fig 5 illustrates the average PSTH and 95% confidence interval of the five categories of responses to LW: 1) firing rate peaked near the projected time of collision (TOC); 2) firing rate showed a valley near TOC; 3) firing rate gradually increased over time; 4) slow firing rate increase with no distinct peak; 5) firing rate remained constant during the looming and increased near the end of the recording, after TOC. S3 Table provides a summary of the distribution of response categories for looming against different visual backgrounds. The distribution of response categories was similar

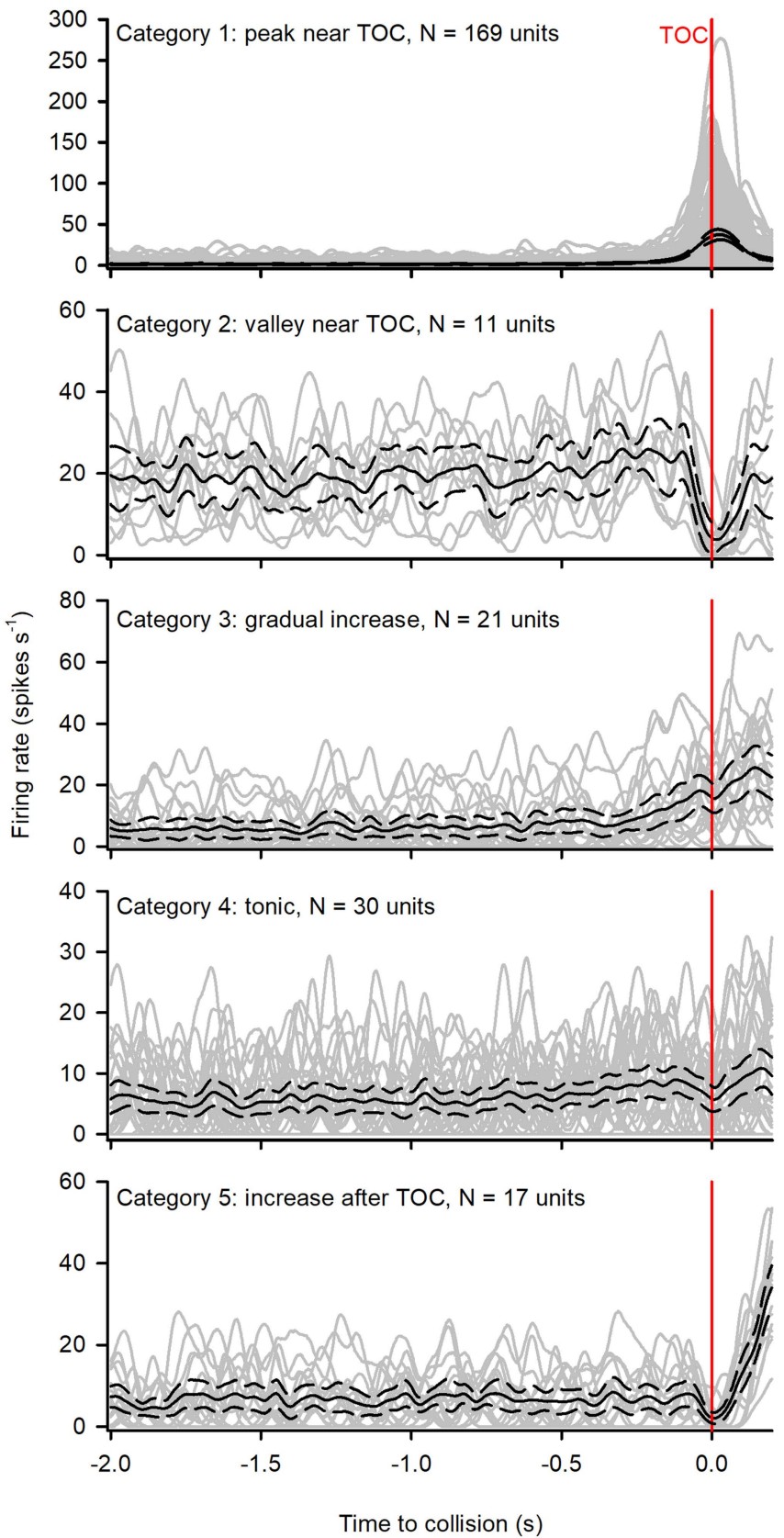

**Fig 5. Categorization of units responding to a looming stimulus against a white background (LW).** All peristimulus time histograms (PSTHs), represented by the grey lines, were aligned to the projected time of collision (TOC), represented by the red vertical line. The black solid lines represent the average PSTH, while the black dotted lines represent the 95% confidence intervals. Category 1 included units that showed a peak near TOC. Category 2 included units that showed a valley near TOC. Category 3 included units that showed a gradual increase during the stimulus presentation. Category 4 included units that showed a slow firing rate increase with no distinct peak. Category 5 included units that showed an increase towards the end of the displayed window, after TOC.

between LW and LG. However, when presented with LF, only 5 units, compared to 21 for LW and 27 for LG, displayed a gradual increase (category 3). Additionally, more units showed a constant firing rate that increased near the end (category 5). It is likely that some units in categories 1 and 3 exhibited a delayed response to LF, leading to their placement in category 5.

Fig 6 shows the responses of units from one locust that were classified into the corresponding response category. In general, units in each category were affected by the background. However, we did not find units from each category in

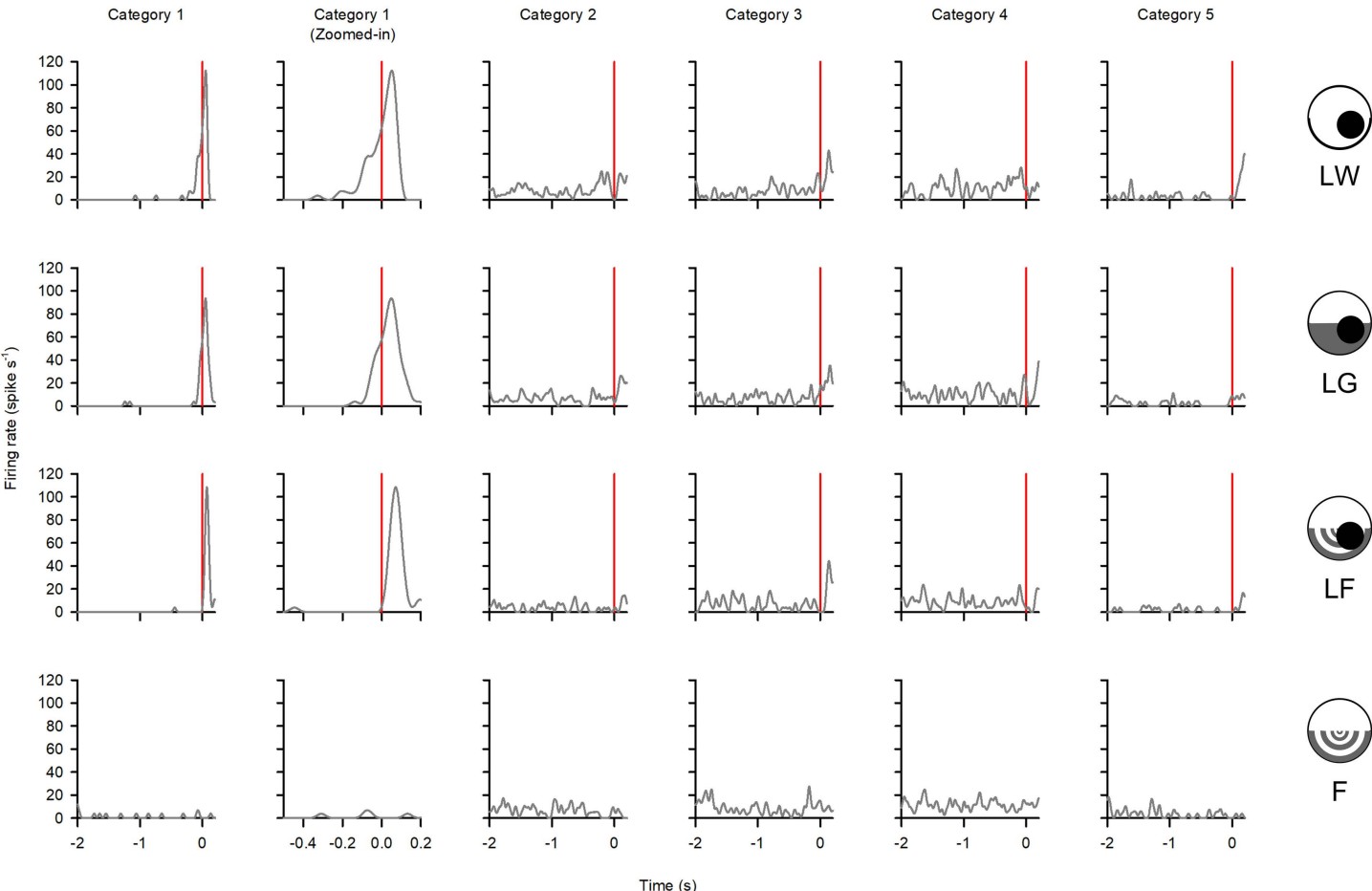

**Fig 6. Responses of spike-sorted units from each response category in one locust.** Peristimulus time histograms show responses of units in each category (columns) to the four stimuli (rows). LW represents a loom against a white background, LG is a loom against a lower grey background, LF is a loom against a lower flow field, and F is a lower field with no approaching object. The red vertical line is the time of collision (TOC). The same time window axes for F were selected after the flow field began moving. The PSTHs of the unit in category 1 was enlarged to show the shape of the peak near TOC.

all locusts, which precludes a detailed analysis of all individual units. Therefore, it was necessary to plot the data pooled from all animals. Recent studies using multichannel recordings revealed different and variable categories of responses to looming stimuli against a white background [42]. Because of the variable responses, there are no a priori response parameters of multiple units responding in the presence of a flow field. Therefore, we did not attempt to measure specific parameters from responses in categories 2–5. Indeed, the intent of this study is to examine responses from putative neural ensembles.

For response category 1, which peaked near TOC, various parameters were calculated, including the peak firing rate, peak time (relative to TOC), rise phase, and decay phase [21,44,45] (Fig 7). Analysis of units in category 1 was based on the resemblance of firing rate modulation properties to those from previous studies that focused specifically on DCMD [20]. The rise phase was defined as the time from when the unit's firing rate last exceeded the upper 95% confidence

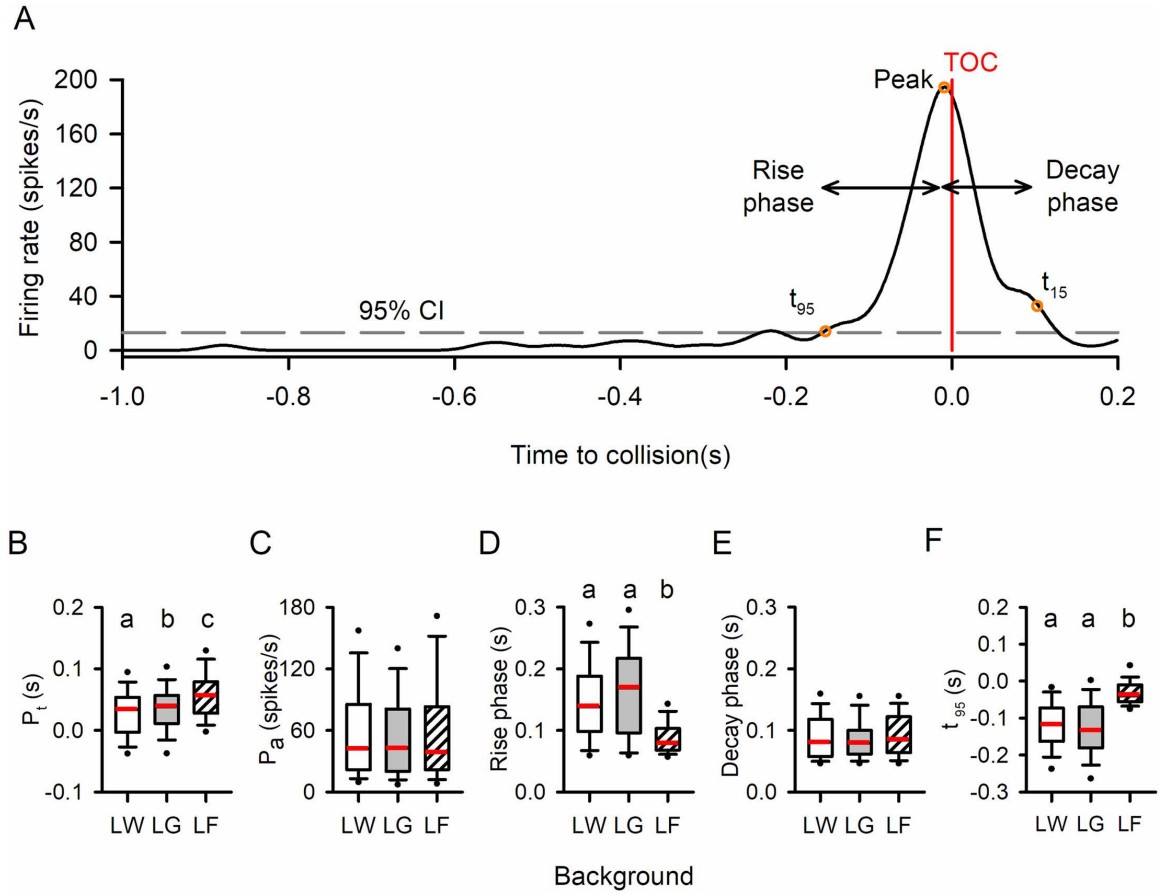

**Fig 7. Comparison of response parameters of units in Category 1 (peak near TOC) between background types. (A)** Sample PSTH of a Category 1 unit in response to a looming stimulus. The red vertical line represents the time of collision (TOC), while the grey dashed line represents the upper 95% confidence interval of the histogram. The last time the instantaneous firing rate exceeded the upper 95% confidence interval ($t_{95}$) was defined as the beginning of the response, while the time when the firing rate decayed below 15% of the peak firing rate ($t_{15}$) was defined as the end of the response. The duration from $t_{95}$ to the peak was defined as the rise phase, and the duration from the peak to $t_{15}$ was defined as the decay phase. **(B)** The peak time relative to TOC ($P_t$), (C) peak firing rate amplitude ($P_a$), (D) rise phase, (E) decay phase, and (F) response start time ($t_{95}$) of each PSTH were measured and compared using one-way repeated measure (RM) ANOVA. Different letters above the boxes represent statistically significant differences. The white and grey backgrounds evoked similar peak times (grey being slightly later), while the flow field evoked a significant peak delay. The flow field also resulted in delayed response initiation and shortened rise phases. N = 135 for B-D and 78 for E. Within each box, the red horizontal line represents the median, while the lower and upper boundaries, the whiskers, and the dots represent the 25/75%, 10/90%, and 5/95% percentiles, respectively.

interval ($t_{95}$), i.e., the response start time, to the peak time ($P_t$), while the decay phase was measured from the peak time to when the firing rate decayed to 15% of the peak firing rate ($t_{15}$). Among the 194 units that responded to looming against all three backgrounds, 135 units (70%) exhibited a category 1 response to all three looming stimuli. Across different backgrounds, there were significant differences in peak time (Friedman Repeated Measures Analysis of Variance on Ranks, $\chi^2_2 = 44.236$, $p < 0.001$), the length of the rise phase (Friedman Repeated Measures Analysis of Variance on Ranks, $\chi^2_2 = 70.673$, $p < 0.001$), and $t_{95}$ (One Way Repeated Measures ANOVA, $F_{2, 134} = 166.393$, $p < 0.001$). However, no significant differences were found in peak firing rate (Friedman Repeated Measures Analysis of Variance on Ranks, $\chi^2_2 = 4.326$, $p = 0.115$) or the length of the decay phase (Friedman Repeated Measures Analysis of Variance on Ranks, $\chi^2_2 = 3.318$, $p = 0.190$). Tukey post-hoc tests indicated that all pairwise comparisons were significantly different for peak time (LW vs LG: q = 4.088, $p < 0.05$; LW vs LF: q = 9.338, $p < 0.05$; LG vs LF: q = 5.250, $p < 0.05$), although the difference in peak time between LW and LG was relatively small. For the rise phase, there was no statistical difference between LW and LG (Tukey post-hoc test, q = 2.711, $p > 0.05$), while both LW and LG showed significantly higher values than LF (LW vs LF: q = 8.650, $p < 0.05$; LG vs LF: q = 11.361, $p < 0.05$). Similarly, for $t_{95}$, there was no statistical difference between LW and LG (Holm-Sidak post-hoc test, $t = 1.166$, $p = 0.245$), while LF showed significantly delayed $t_{95}$ than both LW and LG (LW vs LF: Holm-Sidak post-hoc test, $t = 15.183$, $p < 0.001$; LG vs LF: Holm-Sidak post-hoc test, $t = 16.349$, $p < 0.001$). It is important to note that, as peak times were delayed in LF, the selected trial length (−2 to 0.2 seconds relative to TOC) did not always include a full decay phase, i.e., firing rate did not fall below 15% of the peak firing rate by the end of the trial. Consequently, the sample size for the decay phase comparison (N = 78 units) was smaller than that of the other parameters, making it more difficult to find a significant difference.

In summary, among individual responsive units within response category 1, LW and LG elicited similar responses, while the addition of a flow field background (LF) resulted in delayed peak firing and a shorter rise phase for units in category 1, indicating a delayed and more brief response.

## Common trends

Dynamic factor analysis (DFA) was conducted using the MARSS package in R 4.1.3 to extract common trends from all responsive units. Initial attempts to perform DFA on Gaussian-smoothed 1-ms-binned data resulted in consistent crashes of R due to the model's complexity. Consequently, the firing rate was recalculated using a 50-ms bin width without smoothing. Although temporal resolution was reduced, the 50-ms-binned data captured the dynamic modulation of firing rate while significantly reducing computational complexity by a factor of fifty [44].

For each of the 4 visual stimuli, DFA was iteratively performed with an increasing number of common trends, starting from 1. The Akaike information criterion with a correction for small sample sizes (AICc) was used to monitor the performance of each model. When the AICc began to increase, the iteration was halted, and the model with the lowest AICc was deemed the optimal model. S4 Table shows that the optimal models for LW and LG both included 7 common trends. This consistency with the optimal model for looming at 0° elevation against a white background [30] indicates that neither the looming trajectory (45° upwards) nor the half-white-half-grey background influenced the level of variability in the locust's response to a black looming stimulus. In contrast, despite evoking a similar number of responsive units, the optimal model for LF only contained 5 common trends, suggesting less variation among unit responses. This indicates a more stereotyped and temporally shifted population code in the presence of optic flow.

Fig 8 depicts the activity of all the extracted common trends in response to each stimulus. In response to a looming stimulus, irrespective of background types, most common trends exhibited a positive peak near the time of collision (TOC). Though individual unit categorization included responses that generated a negative peak or no clear peak (Fig 5), most units (~70% - see S3 Table) generated a distinct positive peak near TOC. The data reduction methods would have given more weight to these units when calculating the common trends. Therefore, it is not surprising that most common trends showed a positive peak. Against all three backgrounds, only one common trend did not show a peak, and instead,

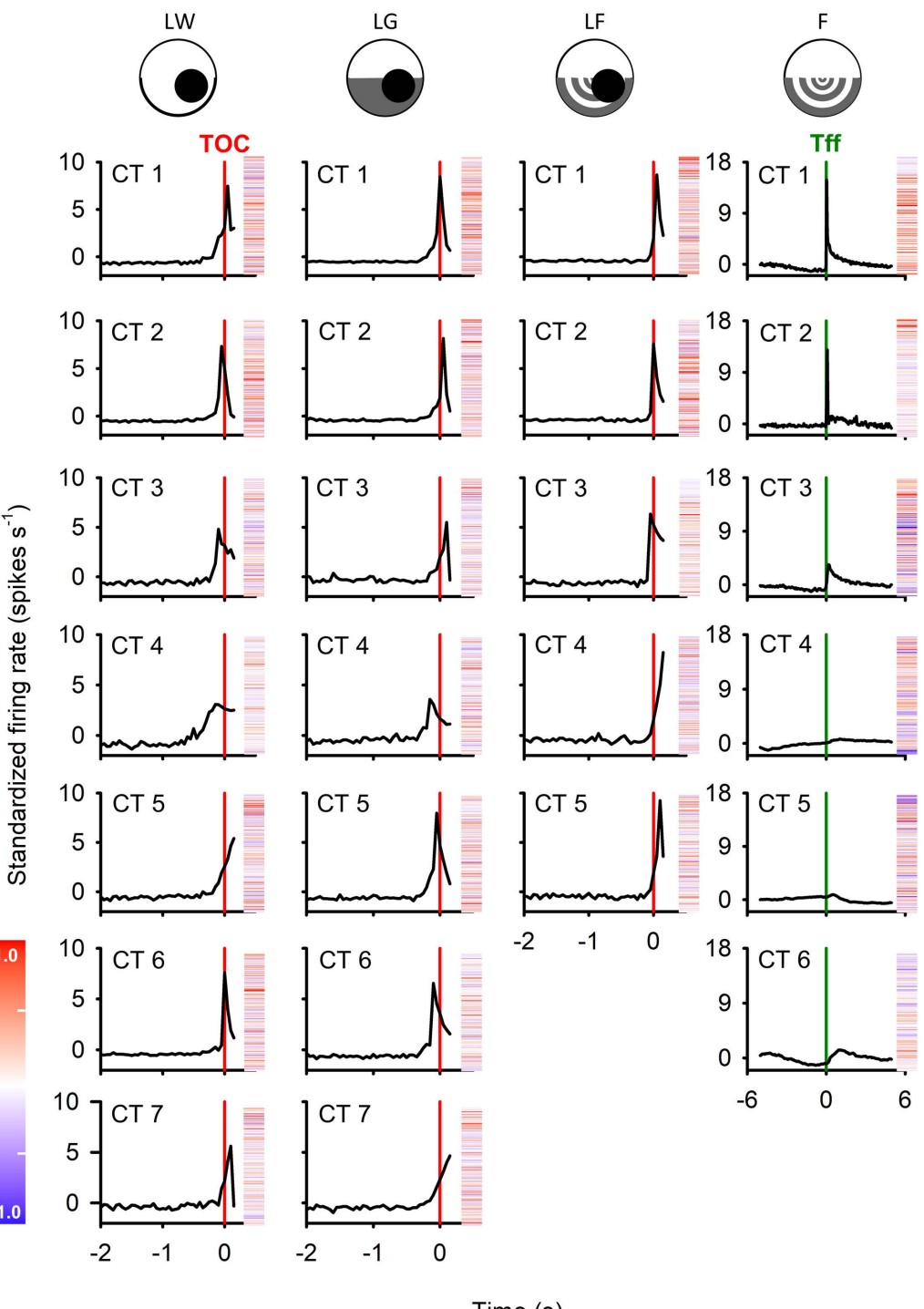

**Fig 8. Common trends (CTs) responses to all four types of visual stimuli.** In each column, peristimulus time histograms (PSTHs) of the CTs from a stimulus type were plotted, aligned to either the projected time of collision (TOC) or the time when the flow field started displaying (Tff). The bin width was 50 ms. In DFA models, the order of CTs was randomized, i.e., CT1 from the first and second columns are not necessarily the same. The factor loadings of constituent spike-sorted units in each CT are shown next to the PSTH in a heatmap. Within a heatmap, each horizontal line represents the correlation of one spike-sorted unit and the corresponding common trend. The order of units in the heatmap is also consistent within a column. Irrespective of background types, most CTs responded to a looming stimulus with a peak near TOC, similar to a DCMD response. The remaining common trend

from each background (CT5 in the white background, CT7 in the half-white-half-grey background, and CT4 in the flow field background) continuously increased until the end of stimulus presentation. For the last visual stimulus, flow field only, CT 1-3 displayed a peak at Tff, when the flow field started, while CT 4-6 remained at a relatively constant level without a clear change near Tff.

continuously increased until the end of the stimulus presentation (CT5 for LW, CT7 LG, and CT4 LF). When presented with the flow field background only (F), without the looming stimulus, the primary type of response was a brief peak right after the time when the flow field started (Tff), followed by a gradual decrease, suggesting adaptation to the flow field. Although the best DFA model contained 6 CTs, the first three CTs all showed this type of response, while the other CTs remained at a relatively stable level with no clear change near Tff.

Among the common trends displaying clear peaks, the peak times occurred before or after TOC in response to LW and LG, while in response to LF, only one common trend (CT 3) had a peak time before TOC, and all other common trends had peaks after TOC. The median peak time of LF common trends was the highest (i.e., latest) among all three looming stimuli, and the rise phase for LF was the shortest. These findings aligned with the peak times observed in category 1 unit responses, although the comparisons of peak time and rise phase were not statistically significant (One-way ANOVA; peak time, $F_2 = 0.507$, $p = 0.618$; rise phase, $F_2 = 0.798$, $p = 0.480$) (Fig 9), likely due to small sample size (N = 4 (LW), 5

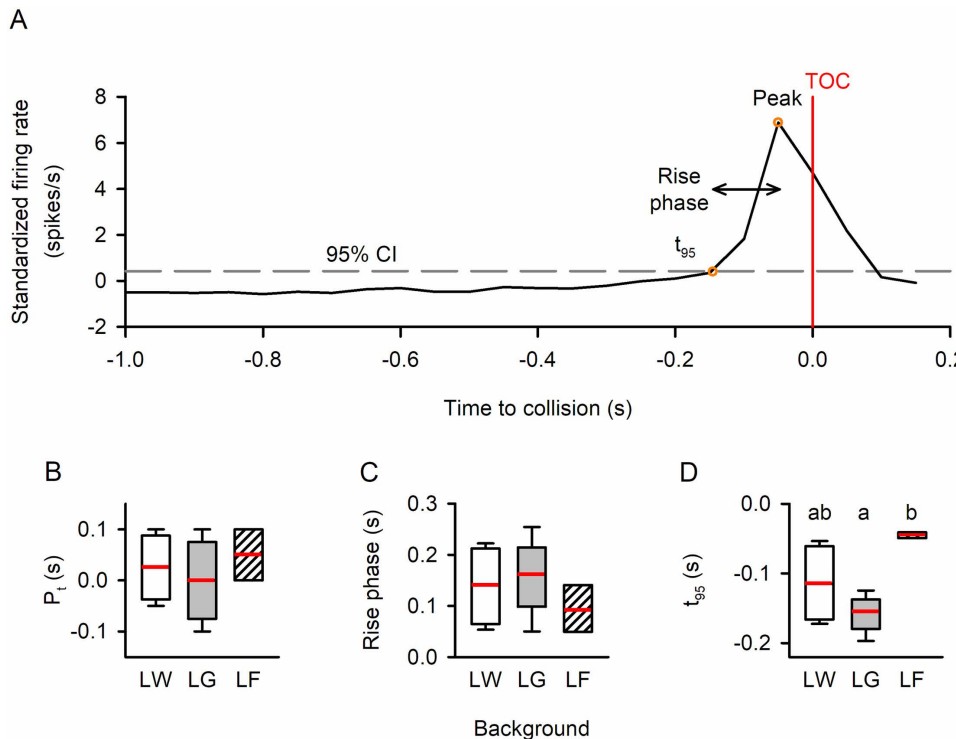

**Fig 9. Comparison of response parameters of common trends (CTs) that peaked near TOC. (A)** Sample PSTH of a common trend in response to a looming stimulus. The red vertical line represents the time of collision (TOC), while the grey dashed line represents the upper 95% confidence interval. The last time the instantaneous firing rate exceeded the upper 95% confidence interval ($t_{95}$) was defined as the start of the response. We defined the duration from $t_{95}$ to the peak as the rise phase. **(B–D)** Box plots comparing CT firing parameters. The peak time relative to collision (B), rise phase (C), and response begin time (D) of each PSTH were measured and compared using one-way repeated measure (RM) ANOVA. Similar to Fig 7, the flow field background resulted in a later peak and shortened rise phase in the CTs, but the difference was not statistically significant. The response start, however, was significantly later in the flow field background, compared to the white and grey backgrounds. Within each box, the red horizontal line represents the median, while the lower and upper boundaries and the whiskers represent the 25/75% and 10/90% percentiles, respectively. Different letters above boxes represent statistically significant differences.

(LG), and 3 (F)) and low statistical power. However, it is worth noting that the response start time ($t_{95}$), defined as when the unit firing rate exceeded the upper 95% confidence interval, was significantly influenced by the type of visual background (Kruskal-Wallis One-Way Analysis of Variance on Ranks, $H_2 = 7.096$, $p = 0.014$). The response began later in response to LF, compared to LG (Dunn's post-hoc test, $p < 0.05$). Since the firing rate did not decay below 15% of the peak firing rate before the end of stimulus presentation in some common trends (e.g., CT 1, 3, and 4), the sample size of $t_{15}$ and decay phase was *too* small (≤ 3 for each background) to derive meaningful comparisons.

## Discussion

During flight, optic flow is a main source of spatial information that an animal can acquire. Components of the optic flow can be used to calculate the distance and trajectory of nearby objects. This is the first study to investigate the effect of a flow field background on looming responses of multiple interneurons in the locust visual system. We found that multiple units were responsive to a looming disc approaching upwards from the bottom right at 45° azimuth and −45° elevation, yet displaying various types of responses. Compared to looming against the white background (LW) and half-white-half-grey background (LG), looming against the flow field background (LF) affected the distribution of different response categories among responsive units. Common trends (CTs) were extracted and compared between background types. Both LW and LG evoked 7 CTs, the same as loom from 90° azimuth at 0° elevation, while LF evoked 5 CTs, suggesting that the flow field background reduced the variability among unit responses. In both individual units and common trends, the flow field background caused the responses to be delayed.

### Technical considerations

The experiments conducted here were, by necessity, in open-loop conditions with recordings from a locust on a fixed tether. Therefore, we were not able to record motor responses to the stimuli, and responses to the motion stimuli did not feed back into the stimulus generation. While it is important to study neural responses and subsequent motor behaviour in closed-loop, it was not the focus of the experiments reported and was beyond the scope of the setup that we used. Nevertheless, this work builds on earlier studies that also used open-loop conditions and, therefore, the data contribute further to our understanding of this system. A modified setup is currently being used to study DCMD responses to object motion against various backgrounds in tethered locusts that can flap their wings and provide feedback to the stimulus parameters in real time.

The number of sorted units that we report here is consistent with previous work using the same semi-automatic sorting method [42]. That study used 8 channels from a 2 x 2 tetrode array positioned along the long axis of the nerve cord. Here, we used a single twisted wire tetrode array that was inserted into the nerve cord, which was supported by a silver hook electrode. Our design provided increased stability to the cord and ensured the placement of all recording sites within the sheath surrounding the cord. Given that the cord between the subesophageal and prothoracic ganglia contains ~60 axons with diameters >5 μm [49], we likely sampled from a high percentage of descending sensory neurons at the recording site.

### Effect of the flow field on collision detection

When an animal moves through its environment, optic flow is generated as a translational or rotational pattern that is perceived by the eye [50]. Animals use optic flow to assess self-motion and stabilize trajectory. For example, honeybees use optic flow to maintain flight height [51], while bumblebees use optic flow in the frontal visual field to control position, flight height, and flight speed [52–54].

In locusts, a flow field consisting of white and grey vertical bars can elicit excitatory postsynaptic potentials (EPSPs) in the LGMD, quickly followed by inhibitory postsynaptic potentials (IPSPs) [45] via activation of lateral and feed-forward inhibition [14,55]. Therefore, LGMD response to small looming objects is inhibited by large-field optic flow, represented by fewer spikes and shorter rise phases [14,20]. In the DCMD, it was found that compared to a white background, a

more complex visual background, such as the flow field, caused reduced peak firing rate, delayed peaks, and shorter rise phases as well [29,37]. We found that flow-field-induced peak time delay and rise phase shortening were common among multiple motion-sensitive neurons, although the peak firing rate was not statistically affected.

The compass cells [56] found in the central complex (CX) of locusts can use polarized light to determine the orientation of the animal [57] and also respond to approaching and translating objects [58,59]. On average, the peak time of the compass cells was delayed in the presence of a flow field. However, the response of most compass cells to looming objects was inhibited by the presence of large-field motion, while some were enhanced [60]. This suggests that a group of neurons is only responsive to looming when optic flow is present, i.e., when the animal is moving. This is similar to recent findings in the monarch butterfly, which show that heading-direction neurons in the central complex change their tuning from sun-bearing coding to a global sun compass during flight [61]. Here we spike-sorted units (n = 25) that responded to LF only, but not to LW or LG (Fig 4). It is possible that these units can enhance motion detection during flight, and play an important role in proper flight manoeuvres within a large swarm.

## Units that responded to the flow field only

We found that a group of units (n = 25) responded to the flow field only, but not to any of the looming stimuli. Despite not directly responding to looming objects, these units can potentially use the flow field to access the self-motion of the animal, and thus factor into the execution of successful collision avoidance behaviours.

Neurons that are exclusively sensitive to large-field motion have been identified in multiple species, such as flies [62], moths [63], praying mantis [64], and crabs [65,66]. In hawkmoths, two types of large-field motion-sensitive neurons, the horizontal cells and the vertical cells, were identified in the lobula plate [67]. In honeybees, horizontal regressive- and progressive motion-sensitive neurons were found to be selectively sensitive to regression or progression motion, respectively [68].

In locusts, although the DCMD does not respond to the initiation of large-field grating stimuli [69], large-field motion-sensitive neurons have been identified in the optic lobe [70–72] and ventral nerve cord [73]. The two lobula directionally selective motion-detecting neurons (LDSMD) respond preferably to flow field moving forwards or backwards, respectively, and one of them synapses with the protocerebral descending directionally selective motion-detecting neuron (PDDSMD) [72,73]. Those neurons found in the medulla, however, did not display directional selectivity [71]. It is worth noting that although the axon of the PDDSMD extends in the ventral nerve cord towards the metathoracic ganglion, it is ipsilateral to the cell body, while we recorded from the contralateral nerve cord. In our study, only one type of flow field was presented, and thus we could not test for directional sensitivity. Based on previous findings, we assume that some of these units, as well as neural ensembles, will display directional preference, while others do not. This can be tested in future studies using various flow field backgrounds and larger-scale multichannel recordings.

## Population coding of motion-sensitive neurons

The response of individual neurons is often "noisy", displaying large variability over repeated stimuli. Compared to individual neurons, functional groups of neurons, i.e., neural ensembles, can represent complicated information and convey robust guides to generate appropriate behavioural responses. In the rat gustatory cortex, neural ensembles progress through reliable and stimulus-specific sequences of states, although the timing of transitions varies between trials [74]. In the locust antennal lobe, multiple projection neurons encode different odorants, while the ensemble activity displays complex patterns over repeated presentation [75]. We found that despite variability among individual neuron responses, the extracted common trends, which represent ensemble activity, remain largely stable. The dominant types of common trends match the response of previously identified motion-sensitive interneurons, providing a reliable interpretation of the given stimuli.

While repeated co-activation of the same neurons can strengthen their synaptic connections and increase the likelihood of the same neural ensemble reoccurring, mechanisms like changes in short-term synaptic dynamics enable flexible reconfiguration of ensemble composition to meet varying computational needs [76]. In the rat anterior cingulate cortex,

ensemble activity during a reward-searching task is affected by previous behavioural history [77]. Physiological states can also determine the level of synchronization among neuron populations [78]. In *Drosophila*, previous experience and hunger state affect the ensemble coding during the determination of the food value [79]. Here, we showed that when presented with identical looming stimuli against different backgrounds, individual unit responses and common trends were both affected. Since the units were generally from the same pool, changes in common trends, both the total number and the peristimulus time histogram, reflect the fluidity of the contribution from individual units. This fluidity could have implications for collision avoidance behaviour. While it is known that the timing of DCMD activity relates to the initiation of avoidance behaviours [25,26], it is not clear how the effects of a flow field on the timing of DCMD spikes would affect the putative triggering of avoidance. While delay in peak firing and change in the decay slope could delay the behaviour, this likely represents an adaptive trade-off. In conditions containing self-generated optic flow, the nervous system may adopt a more conservative triggering policy, delaying responses to prevent false alarms, while producing a more consistent output once activation occurs. Through fluidity of putative neural ensembles, as suggested here, the collision avoidance system could be imparted with the ability to dynamically alter the initiation and magnitude of responses in the face of behavioural activity states and complex visual environments that include optic flow. Therefore, future studies could incorporate closed-loop experiments to determine how individual unit and ensemble responses relate to changes in the animal's motor output when successfully avoiding collision. Our results provide neural grounding for models that treat visual motion cues as drivers of group coordination and selective responses in crowded environments [80–82].

## Supporting information

**S1 Table. Summary of spike detection and unit discrimination.** On average, 27215 spikes (waveforms) were detected, and 16.8 units were spike-sorted. The MANOVA results show that the spike-sorted units were statistically distinct from each other.
(TIF)

**S2 Table. Summary of the number of units that responded to each type of visual stimulus.** LW represents loom against the white background, LG represents loom against the grey background, and LF represents loom against the flow field background.
(TIF)

**S3 Table. Distribution of units in different categories when presented with looming against different visual backgrounds (LW, white; LG, half-white-half-grey; LF, flow field).** The distribution was similar between LW and LG. However, in response to LF, the proportion of Category 3 decreased, while the proportion of Category 5 increased.
(TIF)

**S4 Table. Summary of dynamic factor analysis (DFA) models.** For each stimulus type, the DFA model was performed iteratively, starting with 1 common trend (CT). The Akaike information criterion (AIC) and AIC corrected for small sample size (AICc) of each model are shown above. Since AICc can prevent over-fitting, it was used to determine the best-fit approximating model. For looming against both white and white/grey backgrounds, the best models contained 7 common trends. For looming against the flow field background, the best model contained 5 common trends. For flow field only, the best model contained 6 common trends.
(TIF)

## Author contributions

**Conceptualization:** Sinan Zhang, John R. Gray.

**Data curation:** John R. Gray.

**Formal analysis:** Sinan Zhang.

**Funding acquisition:** John R. Gray.

**Investigation:** Sinan Zhang.

**Methodology:** Sinan Zhang, John R. Gray.

**Project administration:** John R. Gray.

**Resources:** John R. Gray.

**Software:** Sinan Zhang.

**Supervision:** John R. Gray.

**Validation:** Sinan Zhang, John R. Gray.

**Visualization:** Sinan Zhang, John R. Gray.

**Writing – original draft:** Sinan Zhang.

**Writing – review & editing:** Sinan Zhang, John R. Gray.

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
