## [Decision Letter · Decision Letter 0]

24 Apr 2025

Dear Dr. Zhang,

Thank you for submitting your manuscript to PLOS ONE. After careful consideration, we feel that it has merit but does not fully meet PLOS ONE’s publication criteria as it currently stands. Therefore, we invite you to submit a revised version of the manuscript that addresses the points raised during the review process.

While the reviewers have found merit in your work, they raised significant concerns regarding the illustration of the data (e.g. unclear images, not enough raw data), the methodologies employed, data processing steps, etc. Please address all concerns raised by reviewers before resubmitting your manuscript.

We look forward to receiving your revised manuscript.

Kind regards,

Tudor C. Badea, M.D., M.A., Ph.D.

Academic Editor

PLOS ONE

Journal Requirements:

“Natural Sciences and Engineering Research Council of Canada (Award RGPIN-2019-03983)”

3. We note that Figure 1 in your submission contain copyrighted images. All PLOS content is published under the Creative Commons Attribution License (CC BY 4.0), which means that the manuscript, images, and Supporting Information files will be freely available online, and any third party is permitted to access, download, copy, distribute, and use these materials in any way, even commercially, with proper attribution. For more information, see our copyright guidelines: http://journals.plos.org/plosone/s/licenses-and-copyright.

Additional Editor Comments:

While the reviewers have found merit in your work, they raised significant concerns regarding the illustration of the data (e.g. unclear images, not enough raw data), the methodologies employed, data processing steps, etc. Please address all concerns raised by reviewers before resubmitting your manuscript.

Reviewers' comments:

Reviewer's Responses to Questions

**Comments to the Author**

1. Is the manuscript technically sound, and do the data support the conclusions?

Reviewer #1: Yes

Reviewer #2: Yes

Reviewer #3: Partly

2. Has the statistical analysis been performed appropriately and rigorously?

Reviewer #1: Yes

Reviewer #2: Yes

Reviewer #3: No

3. Have the authors made all data underlying the findings in their manuscript fully available?

Reviewer #1: Yes

Reviewer #2: No

Reviewer #3: No

4. Is the manuscript presented in an intelligible fashion and written in standard English?

Reviewer #1: Yes

Reviewer #2: Yes

Reviewer #3: Yes

Reviewer #1: The paper by Zhang et al. represents yet another contribution to our extensive and growing knowledge related to the response of motion-sensitive descending interneurons in the locust. This well-characterized system has successfully served for many years as a model for the study of visual responses in complex, dynamic environments.

In this report, the authors present a statistically extensive analysis of the multi-unit response to looming stimuli over different visual backgrounds, including a flow-field background. The major finding is that, while background complexity has some effect on the temporal aspects of the multi-unit response, the basic looming response remains intact.

While somewhat difficult to follow due to the nature of the work, I find that, overall, the paper is well-written and the findings are well-presented. Below, I list some minor as well as more major comments (in order of appearance in the text rather than importance):

1. L 66: Density dependent phase in locusts is much more than a morphological state.

2. Figure 1: I suggest adding an indication of the direction of motion of the wide-field stimulus background (LF and F) to enhance clarity.

3. L 166: The rationale behind the decision to maintain both the looming stimulus and the flow field within the lower half of the dome screen is not trivial as a locust flying within a dense swarm of conspecifics encounters an extremely complex visual background also in the sky or in its entire visual field. This may thus bother the reader throughout, until it is explained only in the Discussion (paragraph starting at line 514). Consider adding a few words of explanation already at a much earlier point (Intro or Methods).

4. Unfortunately, the low resolution of the figures in the PDF file hampered their review.

5. Figure 4: The figure legend is not clear enough. Please better explain the "1-3 looms" naming of the different columns. Also, what is the meaning of FF and NR?

6. Figure 6B: The peak time appears to be measured relative to the TOC, which would explain the negative values. I suggest clarifying this in the plot axes.

7. 6. The " Common trends " paragraph, starting on line 424 contains many technical details that would be more appropriate in the Methods section.

8. In the Discussion section, the authors explain in length all the details of their findings, including all minor temporal aspects of the multi-unit electrophysiological response. However, what is very much missing is the behavioral significance of all this, i.e., what is the effect of a slight delay in peak amplitude, or a minor change in the decay slope of the firing rate diagram on the behavioral motor response of the locust, on its ability to escape collision or avoid predators!

Reviewer #2: In the manuscript ‘Background optic flow modulates responses of multiple descending interneurons to object motion in locusts’, Zhang and Gray conducted multichannel extracellular recordings from descending neurons of Locusta migratoria. Many of the recorded neurons were sensitive to a looming stimulus that mimics a sudden approach of a large object, e.g., a predator. The present study explicitly tested how the presence of optic flow cues in the background affects the neural response to a looming stimulus presented in the foreground. Optic flow cues are visual stimuli induced through self-motion and therefore omnipresent when an insect flies. In other words, testing the neural response to a looming stimulus in the presence of optic flow cues in the background represents a more naturalistic stimulus scenario and hence makes the study’s rationale highly attractive to a huge neuroscientific community. In addition to minor concerns, I have only a few major concerns.

Unfortunately, the authors presented mostly grouped data, e.g., figure 5, which makes it difficult for the reader to evaluate the results objectively. More specifically, the reader cannot assess how the response characteristics change with different backgrounds within a unit. As all units were stimulated with all four stimuli, the strength of the author’s experimental approach is to show changes in the response profile within a unit. To appreciate the authors’ findings, it is important to present response profiles of single units in response to the four different visual stimuli.

Another major point is that the recordings had been conducted in tethered, non-flying locusts. This means that the presented optic flow cues were uncoupled with the locusts’ state of locomotion (open loop). The authors must at least discuss the study’s technical drawback of recording from a stationary insect. I can imagine that a motor efference copy that closes the loop between the insect’s flight movement and the optic flow cues in the background would make the neurons more sensitive to stimuli that mismatch with the motor efference copy, i.e., a looming stimulus that approaches faster than predicted from the insect’s flight speed.

Major points:

• Line 144-174: At which velocity was the flow field (background) presented? If it was static, it is critical to call it ‘flow’. Based on the introduction, I would suspect to present a looming stimulus in the presence of optic flow that mimics a biologically plausible flight speed of the locust.

• Why did the authors not analyze changes in the response profile induced by different backgrounds for the remaining unit categories (category 2-5)?

Minor points:

Line 52: Please also add a citation of Martin Egelhaaf’s group who studied the influence of optic flow on flight control for decades.

Line 64: But, see the recent work of Marion Sillies who revealed six types of direction-sensitive neurons in Drosophila. (DOI: 10.1126/sciadv.abi7112)

Line 89-91: Please add a sentence about the nature of your used backgrounds. What were you presenting making them ‘naturalistic’?

Line 90: Typo: rigidly-tethered

Line 119: Please specify how the tetrode was fabricated. Which electrodes were used to built the tetrode? Did you measure the electrode resistance before each experiment?

Line 139-140: It might be helpful to add the ‘orientation °’ in Figure 1C. And please increase the size of the schematized locust in the figure.

Line 165-167: Please explain why you decided to present the looming stimulus from below and not from above?

Line 185-197: How large was the time window for spike detection? Maybe you could add a time scale in Figure 2B.

Line 206: Instead of using the term 3D feature space, I prefer using terms like PCA. Especially, when you are using the abbreviation ‘PC’ in the figure which has not been introduced in the legend.

Line 213: ‘not nonresponded’ does not make sense

Line 218: ‘discriminated units’. Terms like ‘single units’ or ‘spike-sorted units’ are more common.

Line 307: A maximal number of 23 recorded units per animal sounds a lot. Also, the mean of 16.8 units per animal is high. I wonder how the entire PCA of one animal looks. In Figure 2, you are only presenting 3 example units that were relatively easily distinguishable by eye. Is this also the case for an entire PCA?

Line 351-366: Category 4 (Tonic response): How does it differ from a non-responsive unit? Is the firing rate in Figure 5 for category 4 units above a background activity (of no visual stimulus)? Unfortunately, I can´t evaluate this based on the figure.

Line 376: How long does it take until the post-TOC response decays for category 5 units? In Figure 5, it is unfortunately clipped.

Line 385-392: Instead of writing that you found significant differences, specify your effect. For example, for the LF condition the units reached their maximal firing rate later but showed a significant shorter rise phase which reflects a more sudden change in the firing rate than in response to looming stimuli presented in absence of a flow field (LW and LG).

Line 399: You should specify the sample size for decay phase in the corresponding subfigure of figure 6.

Line 400: Typo: ‘oof’

Line 424-451: Please specify the rationale for the ‘common trend’ analysis. Can I interpret it as a model-based categorization of different response profiles (instead of the five categories you manually selected beforehand)?

Line 475: Again specify the sample size

Line 522: Typo: ‘bbackground’

Line 522-526: Justifications for the used visual stimuli must be additionally mentioned in the method section. These questions I was asking myself the entire manuscript and had to wait for the answers until the discussion.

Line 532: ‘number of common trends, representing the types of responses among all responsive units’. This info must appear in the result section about the common trends.

Line 539: Missing punctuation mark.

Line 585: ‘In our study’

Line 560-562: In line with this hypothesis, state-dependent central-complex neurons have been recently described in monarch butterflies (doi.org/10.1016/j.cub.2021.11.009).

Figure 2:

• As you are using two different color codes (one for discriminating the signal from each electrode and one for each sorted unit) I needed some time to understand which color stands for what. I suggest keeping the raw Tetrode signal in grayscale and using the color code only for the sorted units.

• Maybe you could rotate the PCA a little so that the distinction between the green and yellow unit becomes clearer. You could also zoom in a little.

• Please color code the ellipses in C so that they can be clearly distinguished from the cumulative response.

• Please indicate from your 20 example units which ones were declared as responsive.

Figure 3:

• For a better comparison across animals, I would plot on the y-axis the percentage of responsive neurons from the recorded neurons rather the absolute number.

• Please add individual data points into your boxplots.

Figure 4:

• Please explain all abbreviations used in the figure in the legend (this comment should be considered for all figures)

• Typo: I assume, on the x-axis it should say ‘F’ instead of ‘FF’?

Figure 5:

• Please indicate the number of neurons presented in each category.

Figure 6:

• Please indicate the number of neurons presented in the boxplots (based on the results I suspect N = 135).

• Please divide the subfigure B into four subfigures (B-E). This makes it much more easier to connect the written results directly to the right subfigure. Also rearrange the boxplot figures so that they fit with the order mentioned in the results (‘peak time’, ‘rise phase’, ‘peak firing rate’, ‘decay phase’).

Reviewer #3: The manuscript describes the analysis of electrophysiological recordings of locust 'looming detector' neurons under semi-naturalistic conditions. The animals are immersed in a type of virtual reality that allows the presentation of sky/ground simulations and a looming object in the shape of a black disc. While the looming detector has been described before, the combination with a non-uniform background is novel and worth investigating. A fairly large number of looming-responsive cells are found, many of which show peak responses in the temporal vicinity of an impending collision. The manuscript is written fairly clearly, and the conclusions seem warranted. However, I have a number of concerns regarding the statistical methods and the data availability -- in its current form, I could not reproduce the results based on the (lack of) data provided.

Major comments:

Data availability:

While the spike detection and sorting seem pretty straightforward, the spike train analyses consitute the most interesting part of the manuscript. To replicate these analyses, the spike train time series would have to be provided -- the summary statistics in the supplements are not sufficient. Especially the more exploratory parts of the analyses, such as the categorization into five response types, are impossible to confirm without access to the spiketrains. They should be provided before the manuscript can be considered for publication.

Methods: please consult with a statisticial on the conditions under which significance tests provide meaningful results (power, test derived from hypothesis rather than post-hoc inspection of data etc.) and use them only under appropriate conditions.

Details:

Intro:

line 89: "...more natural environment.." I agree that the simulated sky/ground is already more natural than previously used stimuli. But why not use the setup for the presentation of an actual sky/ground movie? The setup seems to allow for the necessary resolution etc.

Methods:

line 220: 1 ms bin width, 50ms filter. Why this choice?

line 156: time of collision (TOC). How was that determined, i.e. how large was the disc when 'collision' happened and why is that a sensible choice?

fig 2a: please put units on the time axis.

line 221: "a 2.2-second window..." in fig 2C the window seems to start 1 second before TOC, and last for a fraction of a second afterwards. Why is the rest of the response not shown?

fig 2C: please explain why the interval of 99% confidence (in a constant response?) is elliptic in shape. I understand why the confidence interval shrinks to size 0 at the end of the analysis interval, since then the cs(T) is equal to A (lines 228-230). But why should the confidence interval be equally small at the beginning of the interval?

line 291: "no inherent relationship" please define what an inherent relationship is.

line 308: "Multivariate Analysis of Variance (MANOVA) verified the statistical distinctiveness of these units in three-dimensional cluster space" Please describe the space in which this analysis was carried out, and what the (in)dependent variables were.

350-366: where do the categories come from? If the data were sorted according to visual inspection, then the chi-square test is misleading, because these categories were generated from the data rather than from a hypothesis, so this constitutes 'harking'. Please remove the tests and just describe your visual impression, if that is how you arrived at the five response types.

lines 379-401: please also delete the misleading significance test results and just report the descriptive statistics -- at this point there seems to be no theory/hypothesis from which a statistically testable prediction could be derived?

line 429: "Although temporal resolution was reduced, the 50-ms-binned data captured the dynamic modulation of firing rate while significantly reducing computational complexity by a factor of

fifty." How was it determined that the 50-ms binning captures the rate modulation well?

line 442-443:"In response to a looming stimulus, irrespective of background types, most common trends exhibited a positive peak near the time of collision (TOC)." This seems to contradict the 5 response types depicted in fig 5. E.g. how can a constant response be explained by a positive peak near TOC? Or a valley or a peak after TOC?

line 492: the result of this ANOVA seem to hinge on the (arbitrary) choice of 'response start' (line 488). I recommend describing just the descriptive statistics in figure 8 and deleting the ANOVA results, which do not seem to be derived from a testable hypothesis?

The discussion is adequate and interesting.

**Do you want your identity to be public for this peer review?** For information about this choice, including consent withdrawal, please see our Privacy Policy

Reviewer #1: **Yes: ** Amir Ayali

Reviewer #2: No

Reviewer #3: No

---

## [Author Response · Author response to Decision Letter 1]

3 Jul 2025

We thank the reviewers for their carefully considered and constructive comments on the manuscript. We have made an effort to incorporate all suggestions and edits and provide a response to each comment below in red text.

Reviewer #1:

The paper by Zhang et al. represents yet another contribution to our extensive and growing knowledge related to the response of motion-sensitive descending interneurons in the locust. This well-characterized system has successfully served for many years as a model for the study of visual responses in complex, dynamic environments.

In this report, the authors present a statistically extensive analysis of the multi-unit response to looming stimuli over different visual backgrounds, including a flow-field background. The major finding is that, while background complexity has some effect on the temporal aspects of the multi-unit response, the basic looming response remains intact.

While somewhat difficult to follow due to the nature of the work, I find that, overall, the paper is well-written and the findings are well-presented. Below, I list some minor as well as more major comments (in order of appearance in the text rather than importance):

1. L 66: Density dependent phase in locusts is much more than a morphological state.

We have changed this sentence to more accurately state that migratory locusts are polyphenic.

2. Figure 1: I suggest adding an indication of the direction of motion of the wide-field stimulus background (LF and F) to enhance clarity.

We have modified Figure 1 D to indicate the direction of flow field motion. We have also changed the thickness of the grey arches to indicate that the lines increased in thickness and the flow field expanded. Therefore, we modified the text of the legend of Figure 1 and relevant description of the Visual Stimulus to match figure.

3. L 166: The rationale behind the decision to maintain both the looming stimulus and the flow field within the lower half of the dome screen is not trivial as a locust flying within a dense swarm of conspecifics encounters an extremely complex visual background also in the sky or in its entire visual field. This may thus bother the reader throughout, until it is explained only in the Discussion (paragraph starting at line 514). Consider adding a few words of explanation already at a much earlier point (Intro or Methods).

This is a valid point. Therefore, we moved the relevant section with the discussion to the Visual Stimulus section of the Materials and Methods and modified the citation numbers.

4. Unfortunately, the low resolution of the figures in the PDF file hampered their review.

The low resolution may be due to the download from the server. Our original files that were submitted were high resolution.

5. Figure 4: The figure legend is not clear enough. Please better explain the "1-3 looms" naming of the different columns. Also, what is the meaning of FF and NR?

We have modified the legend and relevant section of the Results to clarify that 1,2, or 3 Looms refers the number of backgrounds that units responded to. FF was a typo. We have changed it to F to indicate flow field only. We have added text to the legend to indicate that NR refers to no response.

6. Figure 6B: The peak time appears to be measured relative to the TOC, which would explain the negative values. I suggest clarifying this in the plot axes.

We have clarified in the figure legend that peak time is relative to TOC. Placing this description on the axis title would be too long for the dimensions of the panel. We have also clarified that the peak firing rate is denoted as peak amplitude (Pa). We also included these descriptions in the relevant sections of the Materials and Methods.

7. 6. The " Common trends " paragraph, starting on line 424 contains many technical details that would be more appropriate in the Methods section.

Respectfully, we disagree. The Materials and methods section describes how dimensionality reduction was performed. The paragraph that the reviewer refers to is the result of that analysis and, therefore, is more appropriate in the results section.

8. In the Discussion section, the authors explain in length all the details of their findings, including all minor temporal aspects of the multi-unit electrophysiological response. However, what is very much missing is the behavioral significance of all this, i.e., what is the effect of a slight delay in peak amplitude, or a minor change in the decay slope of the firing rate diagram on the behavioral motor response of the locust, on its ability to escape collision or avoid predators!

The reviewer raises an interesting point. We have added a paragraph at the end of the discussion to speculate on how optic flow effects on units and common trend responses could influence behaviour.

Reviewer #2:

In the manuscript ‘Background optic flow modulates responses of multiple descending interneurons to object motion in locusts’, Zhang and Gray conducted multichannel extracellular recordings from descending neurons of Locusta migratoria. Many of the recorded neurons were sensitive to a looming stimulus that mimics a sudden approach of a large object, e.g., a predator. The present study explicitly tested how the presence of optic flow cues in the background affects the neural response to a looming stimulus presented in the foreground. Optic flow cues are visual stimuli induced through self-motion and therefore omnipresent when an insect flies. In other words, testing the neural response to a looming stimulus in the presence of optic flow cues in the background represents a more naturalistic stimulus scenario and hence makes the study’s rationale highly attractive to a huge neuroscientific community. In addition to minor concerns, I have only a few major concerns.

Unfortunately, the authors presented mostly grouped data, e.g., figure 5, which makes it difficult for the reader to evaluate the results objectively. More specifically, the reader cannot assess how the response characteristics change with different backgrounds within a unit. As all units were stimulated with all four stimuli, the strength of the author’s experimental approach is to show changes in the response profile within a unit. To appreciate the authors’ findings, it is important to present response profiles of single units in response to the four different visual stimuli.

We have added a new supplementary figure (S1 Figure) that shows responses of units from one locust that fit into each of the 5 response categories described in Figure 5. We have also added text to the relevant section of the results to describe this figure.

Another major point is that the recordings had been conducted in tethered, non-flying locusts. This means that the presented optic flow cues were uncoupled with the locusts’ state of locomotion (open loop). The authors must at least discuss the study’s technical drawback of recording from a stationary insect. I can imagine that a motor efference copy that closes the loop between the insect’s flight movement and the optic flow cues in the background would make the neurons more sensitive to stimuli that mismatch with the motor efference copy, i.e., a looming stimulus that approaches faster than predicted from the insect’s flight speed.

The reviewer is correct and this is a longer term goal to further investigate the role of visual descending neurons in generation of flight escape behaviour. Currently, however, that goal was not the focus of this work and is beyond the scope of the setup we used. Nevertheless, we have added a section to the discussion to address this point.

Major points:

• Line 144-174: At which velocity was the flow field (background) presented? If it was static, it is critical to call it ‘flow’. Based on the introduction, I would suspect to present a looming stimulus in the presence of optic flow that mimics a biologically plausible flight speed of the locust.

We have renamed “circles” to “arches” to better represent their shape. We added text to the Materials and methods that describes the flow field velocity and that it represents forward motion at 3 m s-1, which is the average flight speed of a locust. We have changed the graphic representation of the flow field in Figs. 1, 3, 4, and 7 to demonstrate that the width of the arches increased during expansion.

• Why did the authors not analyze changes in the response profile induced by different backgrounds for the remaining unit categories (category 2-5)?

We have added a section to the results to clarify why we did not measure specific parameters form the other response categories.

Minor points:

Line 52: Please also add a citation of Martin Egelhaaf’s group who studied the influence of optic flow on flight control for decades.

We have added a citation for: 1. Egelhaaf M, Lindemann JP. Path integration and optic flow in flying insects: a review of current evidence. J Comp Physiol A. 2025;211: 375–401. doi:10.1007/s00359-025-01734-9

Line 64: But, see the recent work of Marion Sillies who revealed six types of direction-sensitive neurons in Drosophila. (DOI: 10.1126/sciadv.abi7112)

We have updated this reference.

Line 89-91: Please add a sentence about the nature of your used backgrounds. What were you presenting making them ‘naturalistic’?

We have modified this sentence to to better represent the stimuli we presented. We also added additional text to the materials and methods to further describe the backgrounds.

Line 90: Typo: rigidly-tethered

We have made the change

Line 119: Please specify how the tetrode was fabricated. Which electrodes were used to built the tetrode? Did you measure the electrode resistance before each experiment?

We referenced the tetrode fabrication process that we followed. We have added further text to describe the tetrode impedance.

Line 139-140: It might be helpful to add the ‘orientation °’ in Figure 1C. And please increase the size of the schematized locust in the figure.

We have added this orientation and enlarged the size of the locust. We also added appropriate text to the figure legend.

Line 165-167: Please explain why you decided to present the looming stimulus from below and not from above?

Please see our response to a similar comment from Reviewer 1.

Line 185-197: How large was the time window for spike detection? Maybe you could add a time scale in Figure 2B.

We have added a time scale to Fig. 2B and added details to the relevant section of the text.

Line 206: Instead of using the term 3D feature space, I prefer using terms like PCA. Especially, when you are using the abbreviation ‘PC’ in the figure which has not been introduced in the legend.

We have changed “feature” to “principal component (PC)”.

Line 213: ‘not nonresponded’ does not make sense

This typo has been corrected to read “…not responded…”

Line 218: ‘discriminated units’. Terms like ‘single units’ or ‘spike-sorted units’ are more common.

We have changed “discriminated” to “spike sorted” throughout the document.

Line 307: A maximal number of 23 recorded units per animal sounds a lot. Also, the mean of 16.8 units per animal is high. I wonder how the entire PCA of one animal looks. In Figure 2, you are only presenting 3 example units that were relatively easily distinguishable by eye. Is this also the case for an entire PCA?

We have added text to the discussion to clarify the validity of the number of sorted units. For Figure 2B we have included all 20 units that were sorted from a merged file of 20 stimulus presentations to one locust. We have modified the text of the methods and caption for Figure 2 accordingly.

Line 351-366: Category 4 (Tonic response): How does it differ from a non-responsive unit? Is the firing rate in Figure 5 for category 4 units above a background activity (of no visual stimulus)? Unfortunately, I can´t evaluate this based on the figure.

We only categorized units that showed a statistical response to the stimulus as described in the methods. Therefore, units in category 4 were different from the non-responsive units. We have clarified in the description in the text of the firing rate trends for all categories and more correctly described the slow firing rate increase in category 4. We also added text to clarify that the categories are similar to those reported in previous studies.

Line 376: How long does it take until the post-TOC response decays for category 5 units? In Figure 5, it is unfortunately clipped.

The response for this category would typically decay within 500 ms after TOC. It appears to be clipped in the figure because we used the same time window as with the other categories (200 ms after TOC). We did not display data beyond this window because it is not behviourally relevant to collision avoidance. In the text of the results, we indicated that that for category 5, the firing rate increased after TOC.

Line 385-392: Instead of writing that you found significant differences, specify your effect. For example, for the LF condition the units reached their maximal firing rate later but showed a significant shorter rise phase which reflects a more sudden change in the firing rate than in response to looming stimuli presented in absence of a flow field (LW and LG).

This description is included in the summary paragraph at the end of this section.

Line 399: You should specify the sample size for decay phase in the corresponding subfigure of figure 6.

We have included the sample size in the text and figure caption.

Line 400: Typo: ‘oof’

We have made the correction

Line 424-451: Please specify the rationale for the ‘common trend’ analysis. Can I interpret it as a model-based categorization of different response profiles (instead of the five categories you manually selected beforehand)?

We added a sentence in the Dimensionality Reduction section of materials and methods as well as a reference to rationalize the dynamic factor analysis, which is further described in detail.

Line 475: Again specify the sample size

We have added the sample sizes to the text

Line 522: Typo: ‘bbackground’

We have made the correction

Line 522-526: Justifications for the used visual stimuli must be additionally mentioned in the method section. These questions I was asking myself the entire manuscript and had to wait for the answers until the discussion.

Please see our response to a similar comment from Reviewer 1.

Line 532: ‘number of common trends, representing the types of responses among all responsive units’. This info must appear in the result section about the common trends.

In response to a similar comment by Reviewer 1, we have moved a section of the discussion that described the flow field to the materials and methods. This section also included the line identified above. In the new section within the methods, we clarified that the common trends that are similar for approaches at 0° and -45° are actually from a separate dataset (originally included in a separate PhD thesis chapter) and not part of the results presented in this manuscript. This separate dataset is intended for a separate publication. The statement was made to further validate the flow field setup that required objects to approach from -45° elevation.

Line 539: Missing punctuation mark.

We have made the correction

Line 585: ‘In our study’

We have changed “my” to “our”

Line 560-562: In line with this hypothesis, state-dependent central-complex neurons have been recently described in monarch butterflies (doi.org/10.1016/j.cub.2021.11.009).

We have added this reference to support our suggestion.

Figure 2:

• As you are using two different color codes (one for discriminating the signal from each electrode and one for each sorted unit) I needed some time to understand which color stands for what. I suggest keeping the raw Tetrode signal in grayscale and using the color code only for the sorted units.

We have changed the colour of the raw recording to grey scale and further modified Figure as described in the relevant comment above. The colour code of the sorted units is the same in A and B.

• Maybe you could rotate the PCA a little so that the distinction between the green and yellow unit becomes clearer. You could also zoom in a little.

The new P

---

## [Decision Letter · Decision Letter 1]

13 Aug 2025

Dear Dr. Zhang,

Thank you for submitting your manuscript to PLOS ONE. After careful consideration, we feel that it has merit but does not fully meet PLOS ONE’s publication criteria as it currently stands. Therefore, we invite you to submit a revised version of the manuscript that addresses the points raised during the review process.

**ACADEMIC EDITOR:**

The reviewers comments have come in and they find your manuscript greatly improved. However several outstanding comments regarding data analysis and presentation have been provided. Of particular significance are the statistical handling of data and visualization of spike analysis output. I invite you to address them in the revised version of your manuscript.

We look forward to receiving your revised manuscript.

Kind regards,

Tudor C. Badea, M.D., M.A., Ph.D.

Academic Editor

PLOS ONE

Journal Requirements:

Reviewers' comments:

Reviewer's Responses to Questions

**Comments to the Author**

Reviewer #1: All comments have been addressed

Reviewer #2: (No Response)

Reviewer #3: (No Response)

2. Is the manuscript technically sound, and do the data support the conclusions?

Reviewer #1: Yes

Reviewer #2: Yes

Reviewer #3: Yes

3. Has the statistical analysis been performed appropriately and rigorously?

Reviewer #1: Yes

Reviewer #2: Yes

Reviewer #3: No

4. Have the authors made all data underlying the findings in their manuscript fully available?

Reviewer #1: Yes

Reviewer #2: Yes

Reviewer #3: Yes

5. Is the manuscript presented in an intelligible fashion and written in standard English?

Reviewer #1: Yes

Reviewer #2: Yes

Reviewer #3: Yes

Reviewer #1: All my comments have been adequately addressed, and the paper may be accepted for publication in its current state

Reviewer #2: The manuscript has substantially improved after the revision. My last major concern is about the spike-sorting approach which gets exemplified in figure 2. I appreciate the authors’ transparency by showing all 20 clusters in the PCA but unfortunately, I can`t distinguish the central clusters that are entirely overlapping in the current figure, e.g. magenta, white and yellow cluster. I strongly recommend showing the central clusters in a magnified view, so that the reader can assess the sorting.

Similar problems with quality assessment of spike sorting appear in the spike shapes presented on top of B. Too many clusters are overlapping, making it impossible to see every cluster’s spike shape. In addition to this, colors are replicating, i.e., three clusters are shown in red, three in yellow … . This makes it impossible to assign each cluster to the corresponding spike shape. I therefore strongly recommend splitting the spike-shape images into multiple subimages.

Figure 2: Please increase the size of Fig. 2A, the sorted units (especially the small amplitude ones) are barely visible.

Line 63: After the revision, the term ‘similar to mice’ does not fit anymore as you were previously referring to four types of direction selective neurons in mice

Line 132: Please give some specs about the silver ground wire (diameter, fabricate)

Line 148: To me, the arrows appear yellow

Line 248: What exactly did you statistically compare with the MANOVA? The distance between the clusters in the PCA, e.g., the centroids or the spike shapes between clusters? Please give a few more infos on that.

Fig 3:

“N” for data points is missing in the legend

Line 390: Please specify individual data points (individual locusts?)

Line 434-435: Please place this info to line 421

Category 5: To me it looks like that the firing rate first drops before TOC and increases afterwards

Figure 6:

Please highlight the median of the boxplots in red (especially for the LF condition it is difficult to see the median)

Line 489-503: Please place this passage before line 448. It is more intuitive to read first about the different categories and then getting the explanation why you were focusing on category 1 for detailed analysis.

Line 492-493: Do you suspect all units to be putative DCMD or only category 1 units? This is still unclear as it is written here.

Figure S1: Please place this figure into the main paper. It is the only figure that presents raw data of the effect of the background on the looming response. Please also enlarge the region around TOC so that the reviewer can better see the effect for category 1.

Line 523: Sometimes you write “fig” and sometimes you write “figures”. Please stay consistent.

Figure 7 legend: To get it right: These PSTHs have a binsize of 50 ms? If this is the case please add this info in the legend.

Line 543: What does the color gradient in the heatmap represents?

Is there a reason why you did not compare t95 in figure 6?

Line 576: Similar to figure 6? Not 5

Line 623: Also bumblebees estimate flight height by using optic flow (10.1242/jeb.249763). Please add this reference to your discussion

Reviewer #3: Thank you for providing the data for result replication and additional analyses, I expect this will be useful for the scientific community. I assume the link to the data will be added to the final manuscript, too?

Regarding the clustering analysis: I still strongly recommend removing the misleading chi-square tests, since (line 419ff) "We

identified similar categories based on the trends of firing rate changes over time in response to a similar stimulus ..." i.e. the categories were defined by "eyeballing" the data, which makes significance tests un-applicable and constitutes 'harking'. If these clusters were devised by applying a statistical model to the data (e.g. Gaussian mixtures or similar), then a bayesian model evidence or BIC score could be used to identify the number of clusters.

**Do you want your identity to be public for this peer review?** For information about this choice, including consent withdrawal, please see our Privacy Policy

Reviewer #1: **Yes: ** Amir Ayali

Reviewer #2: **Yes: ** M. Jerome Beetz

Reviewer #3: No

---

## [Author Response · Author response to Decision Letter 2]

22 Sep 2025

We thank the reviewers for their carefully considered and constructive comments on the manuscript. We have made an effort to incorporate all suggestions and edits and provide a response to each comment below.

Reviewer #1: All my comments have been adequately addressed, and the paper may be accepted for publication in its current state

We thank the reviewer for their positive feedback and support for our manuscript.

Reviewer #2: The manuscript has substantially improved after the revision. My last major concern is about the spike-sorting approach which gets exemplified in figure 2. I appreciate the authors’ transparency by showing all 20 clusters in the PCA but unfortunately, I can`t distinguish the central clusters that are entirely overlapping in the current figure, e.g. magenta, white and yellow cluster. I strongly recommend showing the central clusters in a magnified view, so that the reader can assess the sorting.

We have modified Fig 2b to show all unit clusters and enlarged central clusters.

Similar problems with quality assessment of spike sorting appear in the spike shapes presented on top of B. Too many clusters are overlapping, making it impossible to see every cluster’s spike shape. In addition to this, colors are replicating, i.e., three clusters are shown in red, three in yellow … . This makes it impossible to assign each cluster to the corresponding spike shape. I therefore strongly recommend splitting the spike-shape images into multiple subimages.

To address the reviewer's valid concerns, we have reorganized Figure 2. We added a new panel dedicated to the spike shapes of the central clusters, using a distinct and non-repeating color palette to ensure each waveform is clearly attributable to its cluster. We found this approach provides a more readable and effective visualization than a grid-style layout of all 20 units.

Figure 2: Please increase the size of Fig. 2A, the sorted units (especially the small amplitude ones) are barely visible.

We have enlarged the bottom section of Fig 2a. We have also changed the background of the bottom section from black to white, making the colour-coded units easier to see.

Line 63: After the revision, the term ‘similar to mice’ does not fit anymore as you were previously referring to four types of direction selective neurons in mice

Thank you for pointing this out. We have removed the phrase.

Line 132: Please give some specs about the silver ground wire (diameter, fabricate)

Thank you for pointing this out. We have added the required information.

Line 148: To me, the arrows appear yellow

That is a valid point. We have updated the figure legend.

Line 248: What exactly did you statistically compare with the MANOVA? The distance between the clusters in the PCA, e.g., the centroids or the spike shapes between clusters? Please give a few more infos on that.

We used the built-in MANOVA test provided by the Offline Sorter software to statistically validate the separation of our sorted units. This test compares the distributions of spike data points in the 3D feature space, defined by the first three principal components. The independent variable for the test is the assigned unit (cluster), and the dependent variables are the feature vectors (i.e., the PC scores) for each individual waveform within those units. These information have been added to the text too.

Fig 3: “N” for data points is missing in the legend

Thank you for pointing this out. We have updated the info.

Line 390: Please specify individual data points (individual locusts?)

Yes. Individual data points represent different animals. We have updated the info in the figure legend.

Line 434-435: Please place this info to line 421

Thanks for the comment. We have made the change.

Category 5: To me it looks like that the firing rate first drops before TOC and increases afterwards

This is a valid point. However, this drop was also observed in Categories 2, 3, and 4. The distinct feature of Category 5 is the prominent increase after TOC, which resembles some common trends from the DFA (CT5 for LW, CT7 for LG, and CT4 for LF).

Figure 6: Please highlight the median of the boxplots in red (especially for the LF condition it is difficult to see the median)

We have updated figures 3, 6 and 8, highlighting the median line in red. The figure legends have been updated as well.

Update: The previous Fig 6 and 8 are now Fig 7 and 9, respectively, since the previous Fig S1 was moved to Fig 6.

Line 489-503: Please place this passage before line 448. It is more intuitive to read first about the different categories and then getting the explanation why you were focusing on category 1 for detailed analysis.

This is a good point. We have modified the order of corresponding paragraphs to make the flow smoother.

Line 492-493: Do you suspect all units to be putative DCMD or only category 1 units? This is still unclear as it is written here.

Thanks for pointing it out, and the description can indeed cause confusion. Therefore, we have updated the text from “these categories” to “category 1”, since we were only talking about category 1 here.

Figure S1: Please place this figure into the main paper. It is the only figure that presents raw data of the effect of the background on the looming response. Please also enlarge the region around TOC so that the reviewer can better see the effect for category 1.

We appreciate this valuable suggestion. We have moved the figure into the main paper as the new Fig 6. Furthermore, to make the effect for category 1 easier to see, we have added an inset panel that shows a magnified view of the response around the time of collision (TOC).

Line 523: Sometimes you write “fig” and sometimes you write “figures”. Please stay consistent.

We have modified the corresponding texts to ensure consistency.

Figure 7 legend: To get it right: These PSTHs have a binsize of 50 ms? If this is the case please add this info in the legend.

We have added the information in the figure legend.

Update: The previous Fig 7 is now Fig 8, since the previous Fig S1 was moved to Fig 6.

Line 543: What does the color gradient in the heatmap represents?

The heatmap represents the factor loadings of each unit to the specific common trend, i.e., the correlation of each unit to the common trend. We have added further clarification in the Fig 7 figure legend.

Update: The previous Fig 7 is now Fig 8, since the previous Fig S1 was moved to Fig 6.

Is there a reason why you did not compare t95 in figure 6?

Thanks for pointing this out. Initially, we just compared the peak time, rise phase, and decay phase, as routinely done in previous studies. The rise phase was the period between t95 and peak, and we found significant differences in peak time and rise phase already. Since a delayed peak and shortened rise phase would naturally indicate a delayed t95, we did not compare t95. However, since we did compare t95 later in Fig 8, we agree that it is better to include t95 comparison in Fig 6 as well. Therefore, we have included the One-way RM ANOVA comparison of t95 in the corresponding results section, and added a section F in Fig 6 to show the result.

Update: The previous Fig 6 and 8 are now Fig 7 and 9, respectively, since the previous Fig S1 was moved to Fig 6.

Line 576: Similar to figure 6? Not 5

That is correct. Thanks for pointing it out. We have made this edit.

Update: The previous Fig 6 is now Fig 7, since the previous Fig S1 was moved to Fig 6.

Line 623: Also bumblebees estimate flight height by using optic flow (10.1242/jeb.249763). Please add this reference to your discussion

Thanks for suggesting the reference. We have added it to the discussion.

Reviewer #3: Thank you for providing the data for result replication and additional analyses, I expect this will be useful for the scientific community. I assume the link to the data will be added to the final manuscript, too?

Thanks. We will include the link in the final manuscript.

Regarding the clustering analysis: I still strongly recommend removing the misleading chi-square tests, since (line 419ff) "We identified similar categories based on the trends of firing rate changes over time in response to a similar stimulus ..." i.e. the categories were defined by "eyeballing" the data, which makes significance tests un-applicable and constitutes 'harking'. If these clusters were devised by applying a statistical model to the data (e.g. Gaussian mixtures or similar), then a bayesian model evidence or BIC score could be used to identify the number of clusters.

We have removed the chi-squared test, and only kept the description of the distribution of units between different backgrounds.

---

## [Decision Letter · Decision Letter 2]

9 Nov 2025

Dear Dr. Zhang,

Thank you for submitting your manuscript to PLOS ONE. After careful consideration, we feel that it has merit but does not fully meet PLOS ONE’s publication criteria as it currently stands. Therefore, we invite you to submit a revised version of the manuscript that addresses the points raised during the review process.

We look forward to receiving your revised manuscript.

Kind regards,

Teddy Lazebnik

Academic Editor

PLOS ONE

Journal Requirements:

Additional Editor Comments:

All reviewers are positive overall. I invite the authors to address the final comments of the reviewers before acceptance.

Reviewers' comments:

Reviewer's Responses to Questions

**Comments to the Author**

Reviewer #2: All comments have been addressed

Reviewer #4: (No Response)

2. Is the manuscript technically sound, and do the data support the conclusions?

Reviewer #2: (No Response)

Reviewer #4: Yes

3. Has the statistical analysis been performed appropriately and rigorously?

Reviewer #2: (No Response)

Reviewer #4: (No Response)

4. Have the authors made all data underlying the findings in their manuscript fully available?

Reviewer #2: (No Response)

Reviewer #4: Yes

5. Is the manuscript presented in an intelligible fashion and written in standard English?

Reviewer #2: (No Response)

Reviewer #4: Yes

Reviewer #2: Congratulations to the authors who did a good job in revising the manuscript. Thank you for addressing all my concerns and accepting my suggestions. From my side, the manuscript is ready for publication.

Reviewer #4: This is a strong and carefully executed study. The authors examine how background optic flow, simulating self motion, influences neural processing of looming stimuli in locusts. Multi-channel electrophysiology shows that optic flow does not change how many neurons respond, but it shifts response timing later (for both onset and peak) and results in fewer common population trends. The analysis is solid and the work provides valuable insight into how collision detection operates in visually complex conditions. The manuscript is nearly ready for publication and would benefit from the following clarifications and contextual additions.

A. Suggested theoretical context

To strengthen the link between the neurophysiology and real-world behavior, the discussion could acknowledge recent work showing how vision guides collision avoidance and collective movement in locusts. Your results provide neural grounding for models that treat visual motion cues as drivers of group coordination and selective responses in crowded environments.

Consider citing the following works near line 690 or in the final discussion paragraph:

Bleichman I, Yitzhaki O, Bernat NM, et al. 2024. The visual stimuli attributes instrumental for collective-motion tuning in desert locusts. PNAS Nexus. https://doi.org/10.1093/pnasnexus/pgae537

Krongauz DL, Lazebnik T. 2023. Collective evolution learning model for vision based collective motion with collision avoidance. PLOS ONE. https://doi.org/10.1371/journal.pone.0270318

Egelhaaf M, et al. 2023. Optic flow based spatial vision in insects. J Comp Physiol A. https://doi.org/10.1007/s00359-022-01645-z

These papers directly connect visual motion processing to group coordination, collision avoidance and naturalistic movement environments.

B. Clarification in the discussion

In the concluding paragraph, the authors suggest that delayed neural responses in the presence of optic flow may slow behavior or fine tune responses. This can be clarified as a likely adaptive trade-off. When optic flow signals self motion, the nervous system may increase its decision threshold to prevent unnecessary avoidance actions. At the same time, once the threshold is crossed, the reduced variability of population responses suggests a more stable and decisive motor command. Briefly stating this balance between avoiding false alarms and maintaining reliable escape behavior will help tie neural timing and population coding to ecological context.

Suggested phrasing idea:

In conditions containing self-generated optic flow, the system may adopt a more conservative triggering policy, delaying responses to avoid false alarms, while producing a more consistent output once activation occurs.

C. Minor clarity suggestion in Results

In the common trends results, two key findings are presented for the flow condition: fewer common trends and delayed rise time. To improve clarity, consider adding one sentence in the Results or early in the Discussion stating that this indicates a more stereotyped and temporally shifted population code under optic flow. This will help readers follow the logical step from analysis to behavioral interpretation.

**Do you want your identity to be public for this peer review?** For information about this choice, including consent withdrawal, please see our Privacy Policy

Reviewer #2: No

Reviewer #4: No

---

## [Author Response · Author response to Decision Letter 3]

11 Dec 2025

We thank the reviewers for their carefully considered and constructive comments on the manuscript. We have made an effort to incorporate all suggestions and edits and provide a response to each comment below in red text.

Reviewer #2: Congratulations to the authors who did a good job in revising the manuscript. Thank you for addressing all my concerns and accepting my suggestions. From my side, the manuscript is ready for publication.

We thank the reviewer for their positive feedback and support for our manuscript.

Reviewer #4: This is a strong and carefully executed study. The authors examine how background optic flow, simulating self motion, influences neural processing of looming stimuli in locusts. Multi-channel electrophysiology shows that optic flow does not change how many neurons respond, but it shifts response timing later (for both onset and peak) and results in fewer common population trends. The analysis is solid and the work provides valuable insight into how collision detection operates in visually complex conditions. The manuscript is nearly ready for publication and would benefit from the following clarifications and contextual additions.

A. Suggested theoretical context

To strengthen the link between the neurophysiology and real-world behavior, the discussion could acknowledge recent work showing how vision guides collision avoidance and collective movement in locusts. Your results provide neural grounding for models that treat visual motion cues as drivers of group coordination and selective responses in crowded environments.

Consider citing the following works near line 690 or in the final discussion paragraph:

Bleichman I, Yitzhaki O, Bernat NM, et al. 2024. The visual stimuli attributes instrumental for collective-motion tuning in desert locusts. PNAS Nexus. https://doi.org/10.1093/pnasnexus/pgae537

Krongauz DL, Lazebnik T. 2023. Collective evolution learning model for vision based collective motion with collision avoidance. PLOS ONE. https://doi.org/10.1371/journal.pone.0270318

Egelhaaf M, et al. 2023. Optic flow based spatial vision in insects. J Comp Physiol A. https://doi.org/10.1007/s00359-022-01645-z

These papers directly connect visual motion processing to group coordination, collision avoidance and naturalistic movement environments.

We appreciate this suggestion to strengthen the ecological context of our findings. We have added the suggested text and citations to the final paragraph of the Discussion. We explicitly link our results to models that treat visual motion cues as drivers of group coordination and collision avoidance in crowded environments.

B. Clarification in the discussion

In the concluding paragraph, the authors suggest that delayed neural responses in the presence of optic flow may slow behavior or fine tune responses. This can be clarified as a likely adaptive trade-off. When optic flow signals self motion, the nervous system may increase its decision threshold to prevent unnecessary avoidance actions. At the same time, once the threshold is crossed, the reduced variability of population responses suggests a more stable and decisive motor command. Briefly stating this balance between avoiding false alarms and maintaining reliable escape behavior will help tie neural timing and population coding to ecological context.

Suggested phrasing idea:

In conditions containing self-generated optic flow, the system may adopt a more conservative triggering policy, delaying responses to avoid false alarms, while producing a more consistent output once activation occurs.

We agree that framing this as an adaptive trade-off improves the interpretation. We have revised the concluding paragraph of the Discussion to clarify that in conditions containing self-generated optic flow, the system likely adopts a more conservative triggering policy—delaying responses to prevent false alarms while maintaining consistent output once activation occurs.

C. Minor clarity suggestion in Results

In the common trends results, two key findings are presented for the flow condition: fewer common trends and delayed rise time. To improve clarity, consider adding one sentence in the Results or early in the Discussion stating that this indicates a more stereotyped and temporally shifted population code under optic flow. This will help readers follow the logical step from analysis to behavioral interpretation.

We have added a sentence to the "Common Trends" section of the Results to clarify this point. The text now explicitly states that the reduction to 5 common trends indicates a more stereotyped and temporally shifted population code in the presence of optic flow.

---

## [Editor Report · Decision Letter 3]

14 Dec 2025

Background optic flow modulates responses of multiple descending interneurons to object motion in locusts

PONE-D-24-45161R3

Dear Dr. Zhang,

We’re pleased to inform you that your manuscript has been judged scientifically suitable for publication and will be formally accepted for publication once it meets all outstanding technical requirements.

Kind regards,

Teddy Lazebnik

Academic Editor

PLOS One

Additional Editor Comments (optional):

The authors addressed all reviewers' concerns and the manuscript can be accapted now
---

## [Editor Report · Acceptance letter]

PONE-D-24-45161R3

PLOS One

Dear Dr. Zhang,

I'm pleased to inform you that your manuscript has been deemed suitable for publication in PLOS One. Congratulations! Your manuscript is now being handed over to our production team.

Kind regards,

on behalf of

Dr. Teddy Lazebnik

Academic Editor

PLOS One